# Highly active, ultra-low loading single-atom iron catalysts for catalytic transfer hydrogenation

Zhidong An[1,6], Piaoping Yang[2,6], Delong Duan[3,6], Jiang Li [1]✉, Tong Wan[1], Yue Kong[1], Stavros Caratzoulas [2], Shuting Xiang [4], Jiaxing Liu[1], Lei Huang[1], Anatoly I. Frenkel [4], Yuan-Ye Jiang[5], Ran Long [3]✉, Zhenxing Li [1]✉ & Dionisios G. Vlachos [2]✉

Highly effective and selective noble metal-free catalysts attract significant attention. Here, a single-atom iron catalyst is fabricated by saturated adsorption of trace iron onto zeolitic imidazolate framework-8 (ZIF-8) followed by pyrolysis. Its performance toward catalytic transfer hydrogenation of furfural is comparable to state-of-the-art catalysts and up to four orders higher than other Fe catalysts. Isotopic labeling experiments demonstrate an intermolecular hydride transfer mechanism. First principles simulations, spectroscopic calculations and experiments, and kinetic correlations reveal that the synthesis creates pyrrolic Fe(II)-pIN$_3$ as the active center whose flexibility manifested by being pulled out of the plane, enabled by defects, is crucial for collocating the reagents and allowing the chemistry to proceed. The catalyst catalyzes chemoselectively several substrates and possesses a unique trait whereby the chemistry is hindered for more acidic substrates than the hydrogen donors. This work paves the way toward noble-metal free single-atom catalysts for important chemical reactions.

Although heterogeneous catalysts are widely used in industry, their surface heterogeneity and structural complexity make it difficult to decipher the catalytic mechanisms and improve catalyst atom efficiency[1]. Single-atom (SA) catalysts can enable the engineering of the coordination environment of the active center to establish structure-property relationships and elucidate the reaction mechanism[2]. Yet, understanding the actual active center remains difficult despite advances in aberration-corrected transmission electron microscopy, X-ray photoemission spectroscopy (XPS), and extended X-ray absorption fine-structure (EXAFS) due to site heterogeneity and techniques that can unambiguously determine the active site[3].

Recently, SA catalysts have been introduced for biomass conversion[4–22]. Examples include mesoporous N-doped carbon nanofibers anchoring Ru, Pd, and Pt SAs, reaching order of magnitude faster formic acid decomposition than nanoparticle catalysts[4], mesoporous graphitic carbon nitride-supported Ru-SAs for selective hydrogenation or hydrodeoxygenation of vanillin[9], and a chitosan-derived N-doped carbon achieving a high turnover number (TON) of 431 mol$_{phenols}$ mol$_{Ru}^{-1}$ for the reductive catalytic fractionation of lignocellulose[14]. To overcome the bulk reduction of metal oxides by hydrogen, while ensuring high activity and selectivity, Pt SAs anchored onto TiO$_2$ were introduced for the selective C−O bond scission of furfuryl alcohol (FA)

---

[1]College of New Energy and Materials, China University of Petroleum (Beijing), Beijing 102249, China. [2]Department of Chemical and Biomolecular Engineering and Catalysis Center for Energy Innovation, University of Delaware, 221 Academy St., Newark, DE 19716, USA. [3]School of Chemistry and Materials Science, Frontiers Science Center for Planetary Exploration and Emerging Technologies, and National Synchrotron Radiation Laboratory, University of Science and Technology of China, Hefei, Anhui 230026, China. [4]Department of Materials Science and Chemical Engineering, Stony Brook University, Stony Brook, NY 11794, USA. [5]School of Chemistry and Chemical Engineering, Qufu Normal University, Qufu 273165, China. [6]These authors contributed equally: Zhidong An, Piaoping Yang, Delong Duan. ✉e-mail: lijiang@cup.edu.cn; longran@ustc.edu.cn; lizx@cup.edu.cn; vlachos@udel.edu

to 2-methylfuran (MF)[15]. Density functional theory (DFT) calculations and characterization revealed that the cationic redox Pt on $TiO_2$ creates a multifunctional active center that is selective compared to metallic sites. Pt SA supported on the oxygen of defective $Nb_2O_5$ gave >99% MF selectivity at complete conversion due to the synergism of Nb and Pt sites[16], whereby Pt atoms activate $H_2$ and Nb sites activate the C−OH bonds. Most of the aforementioned works have focused on activity invoking noble metal catalysts. Non-noble transition metal-based SA catalysts are appealing owing to their low cost, earth abundance, and high activity[20–22]. For example, $MoS_2$ monolayer sheets decorated with isolated Co atoms, bonding covalently to sulfur vacancies on the basal planes, exhibit superior performance for HDO of 4-methylphenol to toluene[20]. The sulfur vacancies adjacent to the Co-S-Mo sites allow low reaction temperatures (180 °C) without sulfur loss or deactivation. A high loading (7.5 wt%) Ni SA catalyst exhibited high hydrogenation activity of unsaturated substrates and excellent tolerance to harsh conditions due to tight bonding of Ni and N atoms[21]. Lin et al. fabricated a Ni-based SA catalyst on carbon nanotube (CN) ($Ni_{2.1}$/CN) for the CTH of 5-hydroxymethyl furfural (HMF), where the SA $Ni-N_4$ active sites afford a reasonable turnover frequency (TOF) of $22\,h^{-1}$ and chemoselectivity using ethanol as a hydrogen donor[22]. It was hypothesized that pyridinic N of the $Ni-N_4$ site is the active center and electron transfer from N to the Ni lowers the energy barrier for H desorption, facilitating the CTH process.

Furfural (FF) is an unsaturated, commercial biomass derivative with an annual global output of *ca.* 300,000 tons[23]. Its multiple functional groups endow it with diverse pathways to manufacture chemicals and fuels[24] and enable catalyst benchmarking[23]. For instance, CTH of FF to FA with secondary alcohols as hydrogen donors has been highly attractive as a commercially relevant and fundamental probe reaction[25]. As iron is abundant, low cost, and environmentally friendly, we have reported the first study of Fe-catalyzed CTH of FF with 83.0% selectivity to FA and 91.6% conversion at 160 °C in 15 h[26]. Since, other Fe-based catalysts have been reported but usually under harsh conditions. The intrinsic activities of most reported catalysts are low and not accurately measured due to the heterogeneity of Fe-based catalysts[25]. Moreover, the TOF of conventional nanoparticle iron catalysts Fe catalysts is low, $<10\,h^{-1}$.

Here, we successfully prepared SA Fe catalysts using iron nitrates as the Fe precursor via a saturated adsorption strategy onto ZIF-8 at very low Fe loadings (<0.1 wt%). By comparison, classic methods for the preparation of SA Fe/Co catalysts from ZIF-8 usually involved a pore or spatial confinement strategy, where metal complexes with large ligands, such as acetylacetone (acac) or ferrocene, rather than small traditional metal salts, such as nitrates, are necessary[27,28], and $Fe(II)-N_4$ is commonly recommended as the active site. We find the performance of SA Fe catalyst matches state-of-the-art catalysts and its TOF ($1882\,h^{-1}$ at 120 °C and $367\,h^{-1}$ at 80 °C) is two to four orders of magnitude higher than all previous Fe catalysts even at a much lower reaction temperature. The catalyst affords 93.1% yield of FA and 99.5% conversion of FF at 120 °C in only 1 h. Spectroscopic data (XANES and EXAFS) and kinetic experiments point to a single active center, and isotopic labeling experiments indicate intermolecular hydride transfer, following the classic Meerwin-Ponndorf-Verley (MPV) mechanism. First-principles calculations identify $Fe(II)-pIN_3$ as the active site whose mechanism, spectra, and activity are consistent with the experimental data. Notably, we show that the rigid geometry of the $Fe(II)-N_4$ site synthesized in prior work does not allow to simultaneously coordinate the substrate and solvent molecules due to steric hindrance, rendering these sites inactive. We demonstrate a concerted mechanism of IPA binding to the Fe(II) atom and deprotonation on the vicinal N-atom allowing the FF coordination to the Fe(II) atom. The Fe atom relieves its strain associated with the FF binding by being pulled out of the plane of the support. The phenomenon is qualitatively understood by considering the respective crystal field splitting diagrams. Such geometric

effects, manifested in the transition state, have rarely been reported in heterogeneous catalysis. The catalyst is selective for many substrates. Substituent effects demonstrate that additional −OH groups in the substrate reduce the activity.

## Results

### Catalysts synthesis and characterization

The synthesis of iron catalysts using two MOF precursors (ZIF-8 and MOF-5) is illustrated in Fig. 1a. $Fe(NO_3)_3$, a smaller Fe salt, is used as the Fe source, compared to literature's $Fe(acac)_3$[28]. The catalyst prepared using the ZIF-8 precursor after pyrolysis at 800 °C contains ultra-low loading of Fe (<0.1%) vs. 2.16% in Li's paper[28] (Table S1). As Fe coordinates with $H_2BDC$ but not 2-MI, saturated adsorption of $Fe^{3+}$ onto ZIF-8 using 2-MI is recommended to achieve low Fe loading even upon adding excessive Fe precursor[29]. A two-step route is also employed to tune the Fe loading in Fe-ZIF catalysts (Table S2), while similar performance and single-atomic Fe status is achieved as the one-step route at the same Fe loading (Fig. S1 and Tables S3 and S4) (see details in methods section).

TEM images of the ZIF-8 precursor (Fig. S2) and Fe-ZIF-8-800 catalyst (Fig. 1c) reveal a uniform lamellar structure of *ca.* 160 nm ZIF-8 matrix that is well preserved during pyrolysis. Elemental mapping (Fig. 1f) suggests even dispersion of iron in Fe-ZIF-8-800 catalyst, and ICP-AES analysis shows a loading as low as <0.1 wt%. In contrast, pyrolysis of MOF-5 leads to irregular particles (Fig. 1d), indicating the collapse of the initial MOF-5 structure during pyrolysis. Characteristic peaks of ZnO (JCPDS 80−0075) and an extremely weak peak of the (110) facet of metallic iron (JCPDS 87−0721) are observed in the XRD pattern of Fe-MOF-5-800 catalyst. The MOF-5 affords a Fe loading of 5.73% (Table S1). The amorphous carbon peaks in Fe-ZIF-8-800 and ZIF-8-800 and the absence of ZnO and metallic Fe peaks in the XRD patterns of the ZIF-based catalyst (Fig. 1g) suggest that Zn and Fe are in a different state in MOF-5 and ZIF-8 catalysts. Additionally, the Fe-ZIF-8-800 catalyst exhibits a higher level of defects, as indicated by the higher $I_D/I_G$ ratio (Fig. 1h and Table S6), than Fe-MOF-5-800 and ZIF-8-800 catalysts.

The states of Zn and N were investigated via XPS. The slightly higher binding energy of Zn in Fe-ZIF-8-800 than ZIF-8-800 (Fig. 1i) is attributed to electron withdrawal from Zn to Fe even though there are no direct Zn-Fe bonds as shown in the EXAFS spectra (Fig. 2c). That is possible as charge transfer can occur through metal-N bonds, similar to the N-mediated charge transfer in the Cu/Zn-NC catalyst, where direct Cu-Zn coordination was absent according to the EXAFS results[30]. Zn's binding energy in MOF-5 shifts to higher values, correlating to ZnO observed in the XRD pattern (Fig. 1g). Thus, Zn in ZIF-8 may be in $ZnN_4$, as further confirmed by EXAFS[31,32] (Fig. 2f). The fraction of pyrrolic N (the most possible N state in $Fe-N_3$ moiety in Fig. 2g[33–35]) in Fe-ZIF-8-800 is 13.1 at%, higher than Fe-ZIF catalysts pyrolyzed at lower temperatures, and slightly lower than ZIF-8-800 (Fig. 1j and Table S8). Complete loss of Zn and N occurs upon pyrolysis at 1000 °C.

X-ray absorption near-edge structure (XANES) and Fourier transforms (FTs) of EXAFS spectra provide insights into the state of Fe even at low fractions as aberration-corrected transmission electron microscopy (Fig. S8) cannot distinguish Fe and Zn atoms due to approximate atomic number, and XPS is incapable of acquiring significant signals at a Fe loading of <0.1 wt% (Fig. S9)[36]. Fe atoms in the Fe-ZIF-8-800 catalyst are oxidized (Fig. 2a), and the Fe-N length is *ca.* 1.4 Å (Fig. 2c). Trace Fe has limited influence on the oxidation state of Zn, leading to a nearly indistinguishable change in the XANES spectra of Zn K-edge (black and red lines in Fig. 2b). Only one main peak of Zn-N coordination is observed in the ZIF-8 catalysts (Fig. 2d). Wavelet transform of the spectra further confirms that Fe and Zn atoms are well dispersed without aggregation in Fe-ZIF-8-800 and ZIF-8-800 catalysts (Fig. 2i). By comparison, the Fe-Fe and Zn-Zn scatterings are observed in the high *k* value range in Fe and Zn K-space spectra in Fe-MOF-5-800

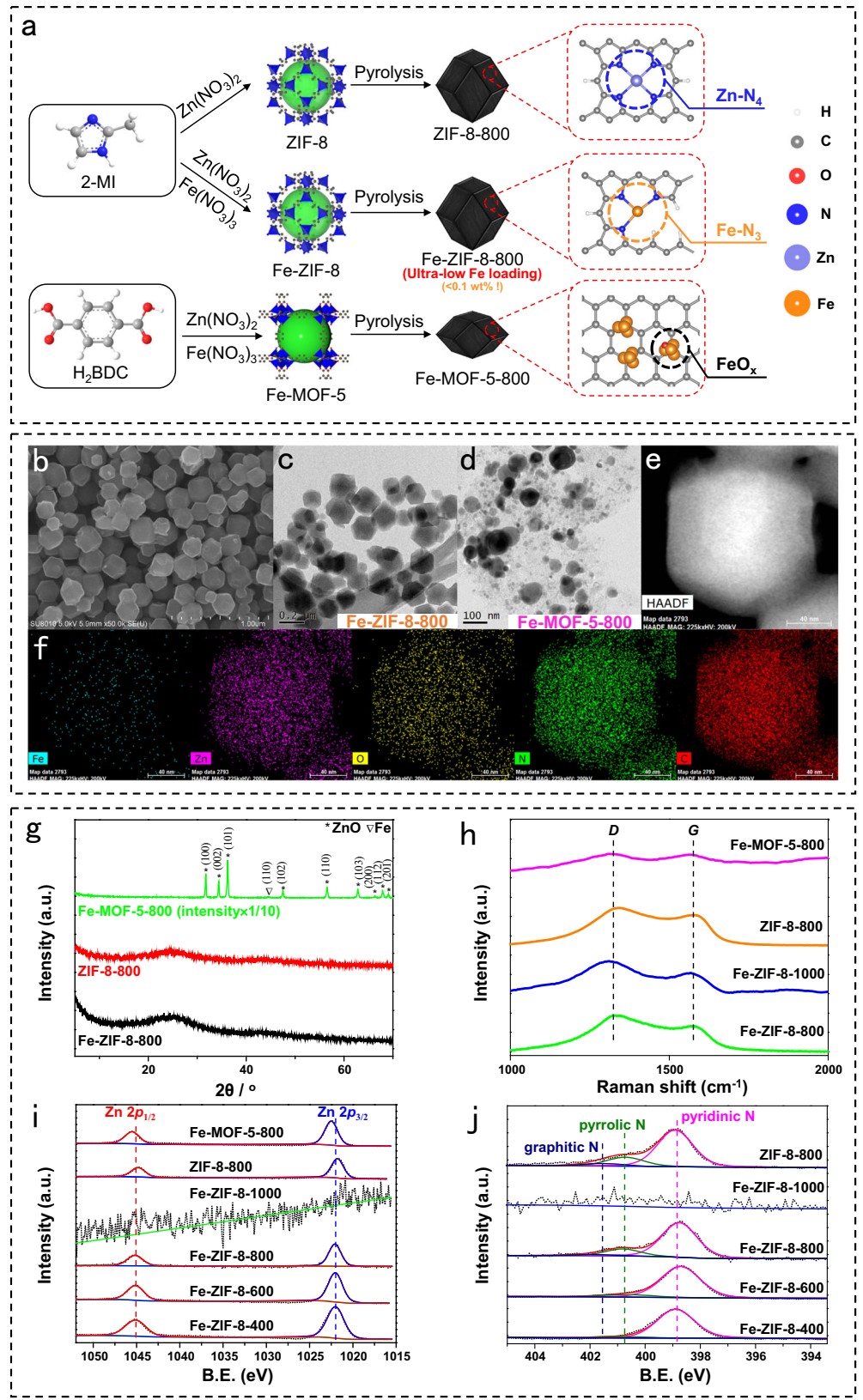

**Fig. 1 | Catalyst synthesis schematic and multiple characterization data.**
**a** Schematic of synthesis of iron catalysts using ZIF-8 and MOF-5 as precursors. The iron loading in Fe-MOF-5-800 catalyst is about 40 times higher than that in Fe-ZIF-8-800 with the same initial amount of Fe(NO₃)₃ (1:13 molar ratio of Fe/Zn). SEM (**b**) and TEM (**c**) images of Fe-ZIF-8-800, and TEM image of Fe-MOF-5-800 (**d**). HAADF-STEM (**e**), and Fe, Zn, O, N, and C elemental maps (**f**) of Fe-ZIF-8-800. **g** XRD pattern of ZIF iron catalysts prepared from different precursors. Raman spectra (**h**) and corresponding Zn 2p (**i**) and N 1s (**j**) XPS spectra of MOF-derived catalysts.

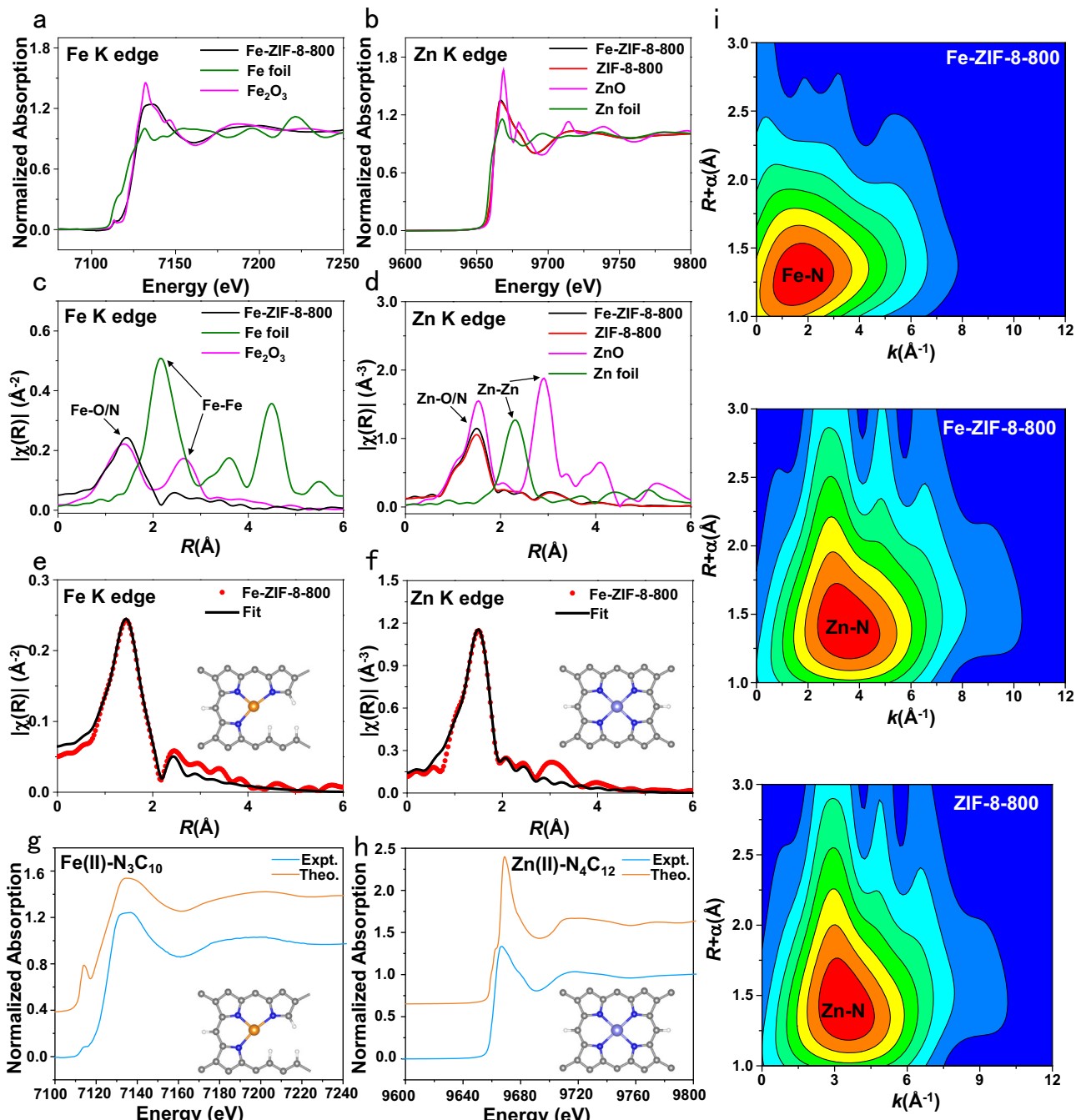

**Fig. 2 | Experimental and simulation X-ray absorption spectra and analysis.** Fe K-edge (**a**) and Zn K-edge (**b**) normalized X-ray absorption near-edge structure (XANES) spectra. Fourier transform of k-weighted Fe K-edge (**c**) and $k^2$-weighted Zn K-edge (**d**) EXAFS spectra. **e, f** EXAFS fitting of the Fe-ZIF-8-800 at Fe and Zn K-edges. **g** Experimental Fe K-edge XANES spectra of Fe-ZIF-8-800 and theoretical spectra of porphyrin-based Fe(II)-N$_3$C$_{10}$ moiety. **h** Experimental Zn K-edge XANES spectra of ZIF-8-800 and theoretical spectra of the porphyrin-based Zn(II)-N$_4$C$_{12}$ moiety. The number in the bracket represents the formal oxidation state of the metal center. **i** Wavelet transform of Fe K-edge EXAFS for Fe-ZIF-8-800 and Zn K-edge EXAFS for Fe-ZIF-8-800 and ZIF-8-800. Insets: atomic structure models of Fe SAs/N-C; Fe (orange), Zn (light blue), N (blue), and C (gray).

catalyst, and the bond distance of Fe-Fe and Zn-Zn bond is *ca.* 2.5 Å and 2.75 Å, respectively (Fig. S10g). The Fe-N coordination number in Fe-ZIF-8-800 catalyst is 3.7 (Table S11), suggesting a mixture of FeN$_3$ and FeN$_4$ structures (Figs. 2e and S11). The coordination number of Zn-N in Fe-ZIF-8-800 is similar to ZIF-8-800 (3.9 vs. 4.1) (Table S12), further confirming the limited influence of trace Fe on the structure of Zn (Fig. 2f).

Computed XANES spectra of metal-N$_x$C$_y$ moieties[33–35] (metal=Fe or Zn) assist identifying the number and type of N ligands coordinated

with the transition metal at the Fe-N$_x$ and Zn-N$_x$ coordination sites in Fe-ZIF-8-800 and ZIF-8-800. The porphyrin-based metal-N$_x$C$_y$ moieties, in which the metal is coordinated to pyrrolic N atoms (plN), e.g., Fe(II)-N$_3$C$_{10}$, Zn(II)-N$_4$C$_{12}$, and Fe(II)-N$_4$C$_{12}$ (Figs. 2g, h, and S12a), reproduce the experimental spectra the best. When pyridinic N is involved in Fe-N coordination (Fig. S13), the edge and post-edge experimental features were not reproduced. Taken together, the EXAFS and XANES analyses suggest that the local structures of the Fe-N$_x$ sites in Fe-ZIF-8-800 are described well by porphyrin-based

moieties Fe(II)-N$_4$C$_{12}$ (Fig. S12a) and Fe(II)-N$_3$C$_{10}$ (Fig. 2g); and the Zn-N$_x$ site of ZIF-8-800 by the Zn(II)-N$_4$C$_{12}$ moiety (Fig. 2h).

## Catalytic performance

Figure 3a shows the catalytic performance and TOF of various catalysts, illustrating the importance of MOF precursors and Fe on reactivity. Some other effects like pyrolysis temperature or H donor were shown in Tables S13 and S14. Strikingly, Fe-ZIF-8-800 catalyst affords 99.6% FF conversion and 96.9% FA selectivity at 120 °C in 6 h. This performance is comparable to state-of-the-art catalysts (Table S15)[26,37–47], significantly better than our previous Fe catalysts, named Fe-phen/C-800[26], and much better than other Fe catalysts under mild conditions (Table S16)[26,48–54]. The Fe-MOF-5-800 and ZIF-8-800 catalysts give lower FF conversion and FA selectivity and TOF of *ca.* 0.054 h$^{-1}$, indicating that the MOF-5 precursor and the absence of Fe lead to inferior catalytic activity, consistent with their poor acid-base properties (Fig. S14). Catalyst precursors without pyrolysis, with or without Fe, give poor FA yields. In an 1 h, only a slight drop in FF conversion and FA yield are achieved.

The TOF of Fe catalysts is benchmarked in Fig. 3a, which are calculated at low FF conversion (<20%) in the kinetic regime[55,56]. Clearly, the TOF of Fe-ZIF-8-800 catalyst (1882 h$^{-1}$, calculated at 120 °C and 5 min with a conversion of 20.5%) is two to four orders of magnitude higher than other Fe-based catalysts. We also calculated the TOF values at various FF conversions and times at 80 °C and 120 °C over the Fe-ZIF-8-800 catalyst (Table S17). The FF conversion correlates well with reaction time at times shorter than 60 min at 80 °C and 120 °C (Fig. S15). The TOF of Fe-ZIF-8-800 catalyst is almost constant vs. time, even at longer times, and two to three orders of magnitude higher than other Fe-based catalysts (Table S16) calculated in the range of 160–200 °C. The excellent activity of the SA Fe underscores that ZIF-derived catalysts have a unique active site.

The effect of the reaction temperature in the range of 80 to 160 °C is shown in Fig. S16. Notably, the Fe-ZIF-8-800 catalyst affords a TOF of 1882 h$^{-1}$ at 120 °C, 206 times higher than that of Fe-phen/C-800 catalyst[26] due to its highly active sites. At 160 °C, a 96.6% yield of FA is obtained in 0.5 h with complete conversion of FF over Fe-ZIF-8-800 catalyst vs. 36.5% yield with 42.9% conversion over Fe-phen/C-800. The major by-products are aldol condensation products of FF and acetone on Fe-ZIF-8-800, also observed over the Fe-phen/C-800 catalyst[26]. At 80 °C in 3 h, the catalytic performance of Fe-ZIF-8-800 at a higher metal loading of 0.54 mol% is comparable to state-of-the-art catalysts (Table S15).

## Reaction mechanism

The linear correlation between Fe loading and specific reaction rate clearly indicates that SA Fe species are the active sites (Fig. 3b). Isotopic labeling experiments confirmed that CTH reaction proceeds via intermolecular hydride transfer, rather than metal-mediated hydrogenation, following the Meerwin-Ponndorf-Verley (MPV) mechanism over Fe-ZIF-8-800 and ZIF-8-800 because of a 1 amu mass spectrometry (MS) shift of the FA formed using 10% 2-propanol-d$_8$ in *t*-butanol solution (Fig. 3c)[57].

To better understand the differences in catalytic performance of Fe-ZIF-8-800 and ZIF-8-800, we performed DFT calculations for the MPV mechanism on periodic models of Fe(II)-plN$_3$, Fe(II)-plN$_4$, and Zn(II)-plN$_4$ (Fig. S17), consisting of Fe(II)-N$_3$C$_{10}$, Fe(II)-N$_4$C$_{12}$, and Zn(II)-N$_4$C$_{12}$ moieties confirmed from the XANES simulations (Fig. 2g, h). The Bader charge analysis (Table S18) confirms that the Fe and Zn atoms carry a positive partial charge and thus can be viewed as Lewis acidic centers.

In a typical MPV reduction of aldehydes over acid-base site pairs of Lewis acid metal-substituted zeolites (e.g., Sn-beta)[58,59], the proton of the alcohol is abstracted by the basic site while the Lewis acid center coordinates the conjugate base of the alcohol (alkoxide ion) and the aldehyde in an octahedral geometry. The ensuing direct hydride transfer from the alkoxide ion to the aldehyde proceeds via a six-member-ring transition state. On Fe(II)-plN$_4$ and Zn(II)-plN$_4$, however, the Fe and Zn atoms cannot coordinate both reactants at the same time due to steric hindrance from the rigid geometry of the metal-N$_4$ site. Nevertheless, the reduction of FF can proceed on both Fe(II)-plN$_4$ and Zn(II)-plN$_4$ by a non-typical MPV mechanism. We have identified two such pathways, one with FF and another with isopropanol (IPA) coordinated to the metal center, respectively referred to as pathway 1 (P1) and pathway 2 (P2) in Fig. 3d. In both cases, the reduction proceeds through a single transition state in which the proton and the α-hydride transfer from IPA to FF in a coordinated manner, leading to FA and acetone (ACE). The optimized geometries of the intermediates and transition states are shown in Figs. S18 and S19. The corresponding Gibbs free energy profiles are shown in Fig. 3f, g. The energy span of P2 (*ca.* 1.4 eV for both Fe(II) and Zn(II)) is somewhat greater than that of P1 (*ca.* 1.2 eV for both Fe(II) and Zn(II)), mainly because when IPA coordinates to the metal, FF physisorbs on the carbon support more strongly than IPA does when the situation is reversed (*viz.* FF coordinated to the metal) but without a commensurate stabilization of the corresponding transition state. Irrespective of the non-typical MPV pathway, however, the similar Fe(II)-plN$_4$ and Zn(II)-plN$_4$ activities are in stark contrast to our experimental data, which show a very low activity of Zn, and suggest that Fe(II)-plN$_4$ might not be the active site responsible for the higher activity of Fe-ZIF-8-800, as shown in Fig. 3a and Table S13.

In contrast, the Fe(II)-plN$_3$ active site model can support a typical MPV mechanism, whereby IPA binding to the Fe(II) atom and deprotonation by a vicinal N-atom is followed by FF coordination to the Fe(II) atom. The FF binding is enabled by the protonation of the N-ligand which weakens the respective Fe-N bond and lets the Fe atom relieve the strain associated with the FF binding by being pulled out of the plane of the support (see optimized geometries in Fig. S20). This pathway (P3) is illustrated in detail in Fig. 3e and the corresponding free energy profile in Fig. 3h. The deprotonation of IPA is facile, requiring 0.28 eV (TS1 in Fig. 3h). The rate-limiting step is the α-hydride transfer from the alkoxide ion to the carbonyl C atom of FF via a six-member-ring transition state (TS2). This step requires 0.87 eV of activation energy (intrinsic) and also determines the overall energy span on this pathway, which is significantly lower than those calculated for the Fe(II)-plN$_4$ and Zn(II)-plN$_4$ models. The reaction is completed by a proton backdonation from the pyrrolic N atoms to the surface furoxy species, FFH$^*$. Microkinetic simulation of the reactions on Fe(II)-plN$_3$, Fe(II)-plN$_4$, and Zn(II)-plN$_4$ showed that the TOF of Fe(II)-plN$_3$ is higher than those of Zn(II)-plN$_4$ and Fe(II)-plN$_4$ by six orders of magnitude (Table S19), suggesting that Fe(II)-plN$_3$ is mostly responsible for the high catalytic activity of Fe-ZIF-8-800 compared to ZIF-8-800.

We note, in passing, that for completeness we also investigated the non-typical MPV pathways P1 and P2 on Fe(II)-plN$_3$. However, the energy spans were 1.19 eV and 1.25 eV for P1 and P2, respectively (Fig. S21), namely, only slightly lower than those on Zn(II)-plN$_4$, which, once again, does not explain the higher activity of Fe-ZIF-8-800 relative to ZIF-8-800.

To gain insights into the higher activity of Fe(II)-plN$_3$ relative to the Fe-site labeled Fe(II)-plN$_4$, we investigated the electronic configurations of the two Fe species. Upon careful analysis, we believe that the two Fe species are in the same oxidation state +2. The DFT magnetic moment of the square planer Fe center in the clean surface of Fe-plN$_4$ is 1.89, indicating a low-spin 3d$^6$ Fe$^{2+}$ configuration. For the Fe-plN$_3$ site, the planar, T-shaped Fe center in the clean surface and the tetrahedral Fe center in the intermediate depIPA_H_FF$^*$ (Fe coordinated to deprotonated IPA and an FF molecule) have magnetic moments of 3.28 μ$_B$ and 3.34 μ$_B$, respectively. These values suggest high-spin 3d$^6$ Fe$^{2+}$ with 4 unpaired electrons or high-spin 3d$^7$ Fe$^{1+}$ with 3 unpaired electrons. However, the 3d$^7$ Fe$^{1+}$ possibility should be discarded

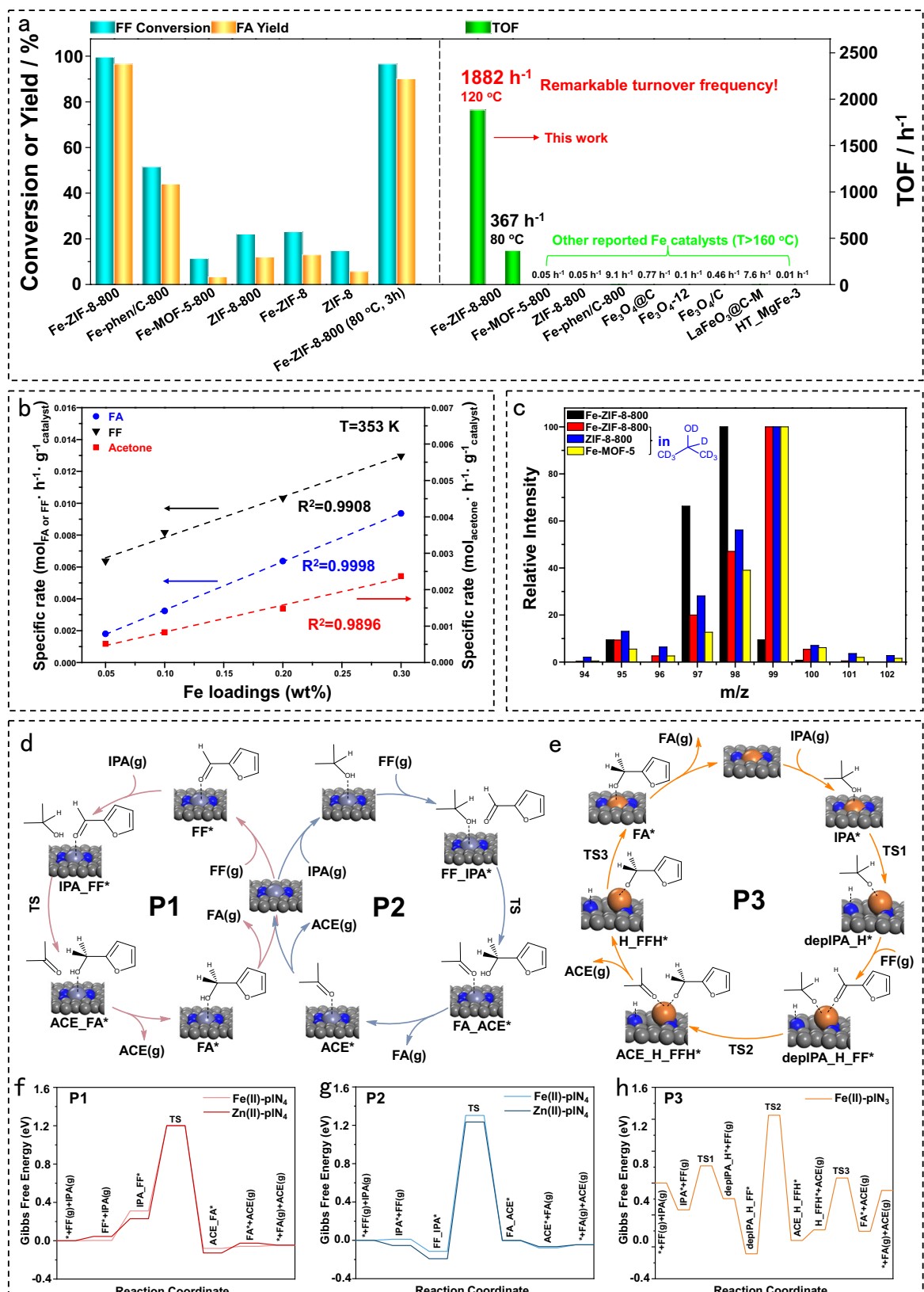

**Fig. 3 | Catalyst performance benchmarking and mechanistic insights from experiments and computations. a** Catalytic performance and TOF of various catalysts. Reaction conditions: 0.5 mmol FF, 3 mL isopropanol solvent, 50 mg catalyst (molar ratio of Fe to FF is 0.13% for Fe-ZIF-8-800 catalyst), 120 °C, 6 h. The iron loading of Fe-ZIF-8-800 at 80 °C and 3 h is 0.3 wt% (0.54 mol%). The details for the calculation of TOFs are listed in Table S16. **b** Correlation of Fe loading with specific reaction rates. Reaction conditions: 0.5 mmol FF, 50 mg Fe-ZIF-8-800 catalyst, 3 mL isopropanol solvent, 80 °C, 5 min. **c** Isotopic labeling experiments. Reaction conditions: 0.5 mmol FF, 50 mg Fe-ZIF-8-800 catalyst, 10% 2-propanol-$d_0$ or $d_8$ in $t$-butanol solution, 120 °C, 3 h. **d** P1 and P2 pathways of CTH reaction on Fe(II)-pIN$_4$ and Zn(II)-pIN$_4$; **e** P3 pathway on Fe(II)-pIN$_3$; **f–h** Corresponding Gibbs free energy profiles of P1, P2, and P3 at 393 K.

because the square planar Fe center in depIPA_H_FF* shows a magnetic moment of 2.07 $\mu_B$, corresponding to only 2 unpaired electrons, namely the low-spin 3d$^6$ Fe$^{2+}$ configuration. Taken together, we conclude that the oxidation state of the Fe center in the Fe-plN$_3$ site is +2, which is the same as that in the Fe-plN$_4$ site. Nevertheless, the two Fe species exhibit different local structures.

The higher activity of Fe(II)-plN$_3$ should be attributed to the anchoring of the Fe atom to a defect site. The resulting three-coordinate metal site in the defective structure promotes the P3 pathway by enabling the FF binding on the Fe(II) center after the deprotonation of IPA. In contrast, Fe(II)-plN$_4$ only supports the relatively unfavorable P1 and P2 pathways because it cannot coordinate both reactants simultaneously. The reasons can be qualitatively understood by considering the respective crystal field splitting diagrams[60,61]. As shown in Fig. S22a, coordination of FF to the Fe(II) center of the trigonal complex depIPA_H* leads to the tetrahedral complex depIPA_H_FF* which is no less stable than depIPA_H* on account of the partially occupied $d_{xy}$, $d_{xz}$ and $d_{yz}$ orbitals lying lower in energy than the $d_{x^2-y^2}$ and $d_{xy}$ orbitals of the trigonal complex. On the other hand, if the square pyramidal Fe(II) center of Fe(II)-plN$_4$ were allowed to bind a second reactant, the uncommon "ridge-tent" looking complex would form, with the two reactants at the ridge over the square planar Fe-N$_4$ complex (Fig. S22b). This would be an unstable complex because the crystal field of this "tent" configuration would be destabilized on account of the four electrons in three orbitals ($d_{z^2}$, $d_{xz}$ and $d_{yz}$) above the barycenter of the field, compared to only one unpaired electron in an orbital ($d_{z^2}$) above the barycenter of the square pyramidal complex with only one of the two reactants as a ligand. This unfavorable electronic configuration precludes coordination of both isopropanol and furfural to Fe(II), inhibiting the P3 pathway.

### Substrate effect

Finally, the effect of the substrate is investigated to demonstrate the versatility of the Fe-ZIF-8-800 catalyst (Table 1). HMF with an additional −CH$_2$OH group gives poor yield of hydrogenated products and low conversion (entry 2). In contrast, a −CH$_3$ on the ring gives a similar conversion but a slightly lower yield to the hydrogenated product (entry 3). The reactivities of aromatic molecules with a phenolic or primary −OH group and a −CHO group (entries 4 and 5) are again inferior. The conversions of p-hydroxybenzaldehyde and p-(hydroxymethyl) benzaldehyde is 18.7% and 44.4%, respectively, affording hydrogenated products with yields of 0.8% and 35.5%, respectively. In contrast, 93.5% yield to benzyl alcohol at 99.8% conversion of benzaldehyde is obtained at identical reaction conditions (entry 7).

In prior work, some of us have seen an electronic effect (either electron withdrawal or donation) of the substituent of the ring on the chemistry[62]. In addition, steric effects, competitive adsorption, and, in general, conformational effects, e.g.[63,64], in adsorption of the substrate arising from different side groups can be at play. Finally, interactions of the functional groups of the ring with the solvent, e.g.[65], can significantly modify the adsorption of the substrate and/or the structure of the solvent around the active site and its ability to carry out the MPV reaction. Entries 5-9 in Table 1 indicate that electronic effects are probably not dominant here. In contrast, the additional −OH group in the substrate (entries 5 and 6) has a significant effect on conversion. The acidity of H atoms in primary and phenolic −OH groups is stronger than in IPA (H donor), and the CTH yields are related to p$K_a$ of the H atoms (Fig. 4, see computation details in Supplementary Note 1 and Supplementary Data 1). These acidic H atoms interact with the active centers of Fe SAs, resulting in inferior CTH reactivities, consistent with the differences in binding energies of −OH groups in HMF and IPA and the adsorption geometries (Fig. S23).

The substitution of the p-position of benzaldehyde with −F and −CF$_3$ groups leads to slightly lower yields of hydrogenated products, while −Cl leads to much lower selectivity (entries 8−10). The CTH

reaction is greatly suppressed with p-substituted t-butyl group, probably due to the steric effect, inhibiting the adsorption of the substrate (entry 11). The reactivity of 2-phenylacetaldehyde is much lower, indicating that the activity of such −CHO group is lower than the benzylic −CHO group (entry 12). In comparison, an unsaturated aldehyde, such as cinnamaldehyde, exhibits better reactivity (entry 13). The replacement of the furan ring with a thiophene ring leads to a slightly lower yield of hydrogenated products, while the pyridine ring results in a much lower product yield (entries 14 and 15). Overall, the catalyst exhibits good chemoselectivity towards CTH of −CHO group without ring hydrogenation but its effectiveness is, as expected, substrate-dependent.

### Catalyst recyclability

Recyclability experiments of the Fe-ZIF-8-800 catalyst (Fig. S24a) show that the CTH activity gradually decreases after the 2$^{nd}$ run, whereas the selectivity to FA is nearly unchanged. The XANES and EXAFS spectra of used Fe-ZIF-8-800 catalyst (Fig. S25) show that the Fe-N coordination number in the spent catalyst is slightly higher than that of the fresh Fe-ZIF-8-800 (4.6 vs. 3.7) (Table S11), while the Zn-N coordination is less changed (4.1 vs. 3.9) (Table S12). After adding additional trace Fe precursor (0.01 wt%) followed by re-pyrolysis, the activity of the spent catalyst can be completely recovered, further confirming that trace Fe is indispensable for the excellent CTH performance of Fe-ZIF-8-800.

## Discussion

Single-atom Fe catalysts have been investigated in various applications and more extensively in electrocatalysis, such as oxygen reduction reaction (ORR). Synthesis of well-defined sites from ZIF-8 has been limited to the pore confinement strategy involving metal complexes with large ligands, giving Fe(II)-N$_4$ pyridinic coordination structures. These have generally been considered as the active sites for the adsorption and reduction of molecular oxygen[28,33,66,67]. Alternative synthesis methods that create different Fe sites have been lacking. Here, the saturated adsorption of Fe(NO$_3$)$_3$ onto the ZIF-8 matrix followed by pyrolysis enables the synthesis of a new Fe SA catalyst with ultra-low Fe loading (<0.1 wt%). Importantly, the Fe catalyst shows a TOF superior to other reported metal and acid-base catalysts for the CTH of FF (Tables S15 and S16).

Our first-principles calculations reveal a profound impact of the structure of the active center on chemistry. The Fe(II)-plN$_4$ and Zn(II)-plN$_4$ sites cannot simultaneously coordinate the substrate and solvent molecules due to steric hindrance from the rigid structure. In contrast, the Fe(II)-plN$_3$ active centers in Fe-ZIF-8-800 allow the co-adsorption of the furoxy species and hydroxyl groups to support a typical MPV mechanism, also confirmed experimentally. The steric hindrance is overcome to a certain extent due to the highly flexible Fe(II)-plN$_3$ coordination structure that partially releases the stress of the Fe atom. Such structural flexibility to enable collocation of reagents and the chemistry to occur has not been reported before for heterogeneous catalysis and enables much faster chemistry. This combination of isotopic labeling experiments and multiscale simulations underscores the importance of catalyst active center structural flexibility and may provide a more general methodology for active site determination.

## Methods
### Chemicals

Iron nitrate nonahydrate (98.5%), alcohols, and N,N-dimethylformamide (99.5%) were purchased from Sinopharm Chemical Reagent Co. Ltd. FF (98.0%). FA (97.0%), 4-fluorobenzaldehyde (98.0%), 4-fluorobenzyl alcohol (98.0%), 4-chlorobenzaldehyde (97.0%), 4-(trifluoromethyl)benzaldehyde (95.0%), 4-[(trifluoromethyl)phenyl]methanol (96.0%), 4-tert-butylbenzaldehyde (95.0%), 4-tert-butylbenzyl alcohol (97.0%), 2-phenethyl alcohol (98.0%), p-phthalic acid (99.0%), benzaldehyde

**Table 1 | CTH of various aldehydes over Fe-ZIF-8-800 catalyst.**[a]

| Entry | Aldehyde substrate | T [°C] | t [h] | Conv. [%] | Product | Yield [%][b] | Sel. [%] |
|---|---|---|---|---|---|---|---|
| 1 | furfural | 80 | 3 | 96.6 | furfuryl alcohol | 90.1 | 93.3 |
|   |   | 120 | 1 | 99.5 |   | 93.1 | 93.5 |
| 2 | 5-hydroxymethylfurfural | 120 | 6 | 20.1 | 2,5-furandimethanol | 13.6 | 67.7 |
| 3 | 5-methylfurfural | 120 | 6 | 100 | (5-methyl-2-furyl)methanol | 79.3 | 79.3 |
| 4 | 4-(hydroxymethyl)benzaldehyde | 120 | 6 | 44.4 | 1,4-benzenedimethanol | 35.5 | 79.9 |
| 5 | 4-hydroxybenzaldehyde | 120 | 6 | 18.7 | 4-hydroxybenzyl alcohol | 0.8 | 4.2 |
| 6 | salicylaldehyde | 120 | 6 | 21.8 | 2-hydroxybenzyl alcohol | 11.0 | 50.5 |
| 7 | benzaldehyde | 120 | 6 | 99.8 | benzyl alcohol | 93.5 | 93.7 |
| 8 | 4-fluorobenzaldehyde | 120 | 6 | 100 | 4-fluorobenzyl alcohol | 89.5 | 89.5 |
|   |   | 160 | 0.5 | 100 |   | 91.4 | 91.4 |
| 9 | 4-chlorobenzaldehyde | 120 | 6 | 36.9 | 4-chlorobenzyl alcohol | 27.6 | 74.8 |
|   |   | 160 | 6 | 99.8 |   | 64.7 | 64.8 |
| 10 | 4-(trifluoromethyl)benzaldehyde | 120 | 6 | 97.9 | 4-(trifluoromethyl)benzyl alcohol | 88.0 | 89.9 |
| 11 | 4-tert-butylbenzaldehyde | 160 | 6 | 25.9 | 4-tert-butylbenzyl alcohol | 22.1 | 85.3 |
| 12 | phenylacetaldehyde | 140 | 6 | 55.7 | 2-phenylethanol | 13.9 | 25.0 |
|   |   | 160 | 6 | 87.5 |   | 17.0 | 19.4 |
| 13 | cinnamaldehyde | 140 | 6 | 52.1 | cinnamyl alcohol | 39.1 | 75.0 |
|   |   | 160 | 6 | 85.5 |   | 62.4 | 73.0 |
| 14 | 2-thiophenecarboxaldehyde | 120 | 6 | 97.9 | 2-thiophenemethanol | 88.0 | 89.9 |
|   |   | 160 | 1 | 100 |   | 87.7 | 87.7 |
| 15 | pyridine-2-carboxaldehyde | 160 | 6 | 90.5 | 2-pyridinemethanol | 69.2 | 76.5 |

[a]Reaction conditions: 0.5 mmol substrate, 3 mL IPA, 50 mg Fe-ZIF-8-800 catalyst.
[b]GC yield.

(98.0%), benzyl alcohol (99.0%), and 4-(2-furyl)-3-buten-2-one (98.0%) were purchased from TCI. 2-methylimidazole (98.0%), 5-hydroxymethylfurfural (99.0%), 2,5-furandimethanol (98.0%), (5-methyl-2-furyl)methanol (97.0%), cinnamaldehyde (98.0%), cinnamic alcohol (98.0%), 2-thiophenecarboxaldehyde (98.0%), 2-thiophenemethanol (98.0%), pyridine-2-carboxaldehyde (98.0%), 2-pyridinemethanol (98.0%), salicylaldehyde (98.0%), and 2-hydroxybenzyl alcohol (98.0%) were purchased from Aladdin Reagent Co. Ltd. Zinc nitrate hexahydrate, phenylacetaldehyde (95.0%), and 5-methyl furfural (99.0%) were purchased from Xiya Reagent Co. Ltd.

## Synthesis of ZIF-8-derived iron catalyst

Catalysts were prepared by pyrolysis of Fe-ZIF-8 precursors under Ar at 400–1000 °C. The prepared catalysts are denoted as Fe-ZIF-8-T, where T represents the pyrolysis temperature in Celsius. All Fe-ZIF-8 catalysts were synthesized using the following procedure. First, zinc nitrate hexahydrate (2790 mg) was added into methanol (100 mL). After the precursor was completely dissolved, 2-methylimidazole (3080 mg) was added to the above solution, followed by the addition of 100 mL of methanol. The mixture was vigorously stirred at room temperature until a white solution was formed, and then 175 mg iron (III) nitrate non-ahydrate was added. The combined mixture was stirred for another 6 h at room temperature. Afterwards, the produced solids were centrifuged at 8260 g for 5 min and washed three times with ethanol before being dried overnight in a vacuum at 60 °C. Finally, the mixture was grinded into a fine powder and then pyrolyzed in a tubular furnace under an argon flow rate of 100 mL min⁻¹. To prepare the best Fe-ZIF-8 catalyst, the temperature program was as follows: 20 °C hold for 60 min, ramp 5 °C min⁻¹ to 800 °C and hold for 2 h. The catalysts prepared from different batches exhibited high catalytic reproducibility.

To tune the Fe loading, a two-step route is adopted. The ZIF-8 precursor was prepared at first, and then being pyrolyzed at 800 °C to furnish ZIF-8-800 support. Certain amount of iron (III) nitrate non-ahydrate was then dissolved in 0.5 mL methanol, and then dropped onto 300 mg ZIF-8-800 support. The mixture was finally pyrolyzed at 800 °C using similar program as the one-step route except another 70 °C hold for 1 h to remove the methanol.

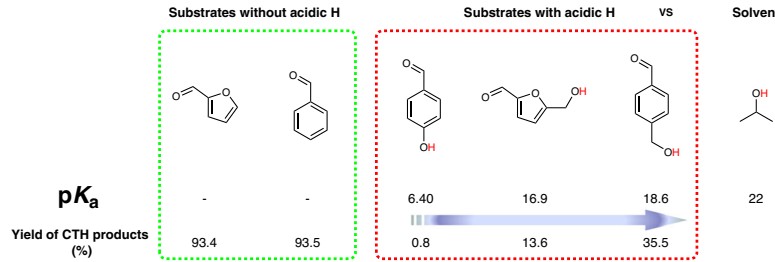

**Fig. 4 | Effect of acidic H atoms in the substrate on CTH of aldehydes over Fe-ZIF-8-800 catalyst.** The p$K_a$ of hydroxyl hydrogen atoms in three different substrates were calculated, and then correlated with their CTH yields.

## Synthesis of Fe-MOF-5-800 catalyst

A typical procedure for Fe-MOF-5-800 catalyst preparation is as follows: First, zinc nitrate hexahydrate (3.63 g) was added into *N,N*-dimethylformamide (100 mL). After the precursor was completely dissolved, H$_2$BDC (3080 mg) was added to the above solution. The solution was vigorously stirred at room temperature for 30 min, and then vigorously stirred at 100 °C until a white mixture formed. Then, 126.3 mg iron(III) nitrate nonahydrate was added. This combined mixture was stirred for 24 h at 100 °C. Afterwards, the produced solids were centrifuged at 8260 g for 5 min and washed three times with ethanol before being dried overnight in vacuum at 60 °C. Then the mixture was grinded into a fine powder and pyrolyzed using the same program as for Fe-ZIF-8-800 catalyst.

## XANES simulations

The Fe and Zn K-edge XANES simulations were performed using the FDMNES code in the framework of multiple-scattering scheme using the muffin-tin approximation for the potential[68,69]. The Hedin-Lundqvist exchange-correlation potential was used in self-consistent calculations. The cluster radius was set to 5.0 Å away from the metal center, with satisfactory convergence being achieved. The calculation results were convoluted by an arctangent function to obtain the final spectrum. The theoretical spectra of reference compounds, Fe foil, FeO, Zn foil, and ZnO, were first simulated and compared with the corresponding experimental spectra to validate the performance of FDMNES (Fig. S26); the convolution parameters (cutting energy and core-level width) for the reference compounds with different formal oxidation states were optimized to obtain the best agreement between experimental and theoretical spectra (Table S22) and applied to the simulations of metal-N$_x$C$_y$ moieties with corresponding formal oxidation states of metal centers. For instance, the cutting energy of −4 eV and core-level width of 2 eV, the optimal parameters of the theoretical spectra of FeO, were used for the spectra simulation of Fe(II)-N$_4$C$_{12}$ moiety. To avoid artificial biases, all models were first optimized by DFT calculations.

## DFT calculations and microkinetic modeling

Spin-polarized DFT calculations were performed using the Vienna ab initio software package (VASP, version 5.4.1)[70]. The electron exchange and correlation effects were described using the optPBE-vdW exchange-correlation functional[71]. The core electrons were represented with the projector augmented wave (PAW)[72] method, and a plane-wave cutoff of 400 eV was used for the valence electrons. The Gaussian smearing method with a smearing width of 0.05 eV was employed. The supercells of Fe(II)-plN$_3$, Fe(II)-plN$_4$, and Zn(II)-plN$_4$ were built based on 3×7 rectangular unit cells of graphene and have the sizes of 12.8 Å×17.2 Å. The vacuum space in the z-direction was set at 25 Å to prevent the interaction between periodic images. The Brillouin zone was sampled with a (3×2×1) k-point grid. All geometry optimizations were performed using the conjugate gradient algorithm. The atomic force convergence of 0.02 eV/Å and the energy tolerance of 10$^{-6}$ eV were employed. The total energies of the gases

were calculated in boxes of 20 Å×21 Å×22 Å using gamma point. The vibrational frequencies were computed within the harmonic oscillator approximation via diagonalization of the Hessian matrix using the central difference approximation with a displacement of 0.015 Å. Transition states were computed using nudged elastic band and dimer calculations[73,74] and confirmed by vibrational frequency calculations. Thermochemical parameters of gaseous species were taken from the Burcat database[75]. The free energies of surface species were corrected using the python Multiscale Thermodynamic Toolbox (pMuTT)[76]. Microkinetic modeling (MKM) was performed using Chemkin[77] in a plug-flow reactor. The MKM parameters are shown in Tables S23 and S24. All reaction pathways, namely, P3 on Fe(II)-plN$_3$, and P1 and P2 on Fe(II)-plN$_4$ and Zn(II)-plN$_4$, were included in the MKM simulations.

## Characterization

The morphology of the catalysts was characterized using a Hitachi SU8010 scanning electron microscopy (SEM, Japan) at 20 kV. Transmission electron microscopy (TEM), HAADF-STEM, and elemental analysis using an energy dispersive spectrometer (EDS) were performed on a JEOL JEM-2100F instrument operated at 200 kV. N$_2$ adsorption measurements were performed on an ASAP2020M adsorption analyzer. XRD was performed at room temperature on an X-ray diffractometer (TTR-III, Rigaku Corp., Japan) with Cu Kα radiation (λ = 1.54056 Å). The data was recorded at 2θ ranges of 5–70°. X-ray photoelectron spectroscopy (XPS) measurements were conducted on an XPS instrument (ESCALAB 250Xi, Thermo-VG Scientific, USA) with monochromatic Al Kα radiation (1486.92 eV). The spectra were fitted using mixed Gaussian-Lorentzian component profiles after subtraction of a Shirley background. The FWHM was fixed as 1.6 eV. The nitrogen content was determined by elemental analysis (Eurovector EA 3000). The iron and zinc content were determined using ICP-AES (Optima 7000DV, PerkinElmer Inc.). NH$_3$-TPD was performed as follows: firstly, approximately 100 mg sample was loaded in a quartz reactor and then heated at 500 °C under argon flow for 2 h. Then the adsorption of NH$_3$ was carried out at 40 °C for 1 h. Subsequently, the catalysts were flushed with argon at 40 °C for 1 h, and then heated to 700 or 800 °C with a ramp rate of 10 °C min$^{-1}$. The procedure of CO$_2$-TPD was the same as NH$_3$-TPD except for the adsorption of CO$_2$ at 40 °C for 1 h. Raman spectra were obtained through a Raman spectrometer (HORIBA Jobin Yvon, France) with λ=532 nm. Thermogravimetric analysis was accomplished using a TG-DTA analyzer Shimadzu DTA-60 instrument. Under Ar flow, a sample was heated from 30 °C to 1000 °C with a ramp rate of 10 °C min$^{-1}$.

## Catalytic transfer hydrogenation of FF to FA

The CTH of FF was carried out using a thick-wall glass tube (15 mL capacity, Synthware). For a typical procedure, FF (0.5 mmol), heterogeneous catalyst (50 mg), and 2-propanol (3 mL) were mixed in a glass tube and then the mixture was vigorously stirred in a silicon oil bath at 120 °C for 6 h. After the reaction, the liquid products were analyzed using both gas chromatography (GC) and gas chromatography-mass

spectrometry (GC-MS). GC-MS analysis was conducted with an Agilent 7890B GC equipped with a DB-WAX 30 m×0.25 mm×0.25 μm capillary column (Agilent). The GC was directly interfaced to an Agilent 5977 mass selective detector (EI, 70 eV). The following GC oven temperature programs were used: 40 °C hold for 1 min, ramp 5 °C min$^{-1}$ to 120 °C, ramp 10 °C min$^{-1}$ to 240 °C, and hold for 5 min. A typical GC spectrum was shown in Fig. S27.

Some experiments were performed at least twice to ensure reproducibility. The carbon loss is attributed to undetected products in GC or coke formation. The conversion and yield for the hydrogenation of FF were calculated in mol%, and TOF was calculated in h$^{-1}$ as follows:

$$\text{Conversion} = \left(1 - \frac{molar\ amount\ of\ FF\ after\ reaction}{molar\ amount\ of\ FF\ before\ reaction}\right) \times 100\% \tag{1}$$

$$\text{Yield} = \frac{molar\ amount\ of\ FA\ after\ reaction}{molar\ amount\ of\ FF\ before\ reaction} \times 100\% \tag{2}$$

$$\text{TOF} = \frac{molar\ amount\ of\ FF\ consumed\ at\ low\ conversions}{molar\ amount\ of\ Fe\ in\ catalysts \times reaction\ time} \times 100\% \tag{3}$$

The stability of Fe-ZIF-8-800 catalyst was investigated in five consecutive runs. The used catalyst was separated from the resulting reaction mixture via centrifugation, followed by three times washes with ethanol. After being dried overnight in vacuum at 60 °C, the catalyst was used directly in the next run. To recover the activity of the used Fe-ZIF-8-800 catalyst, 0.083 mL methanol solution containing 0.036 mg iron (III) nitrate nonahydrate was dropped onto 50 mg used Fe-ZIF-8-800 catalyst. In this case, the Fe dosage is only 0.01 wt%. The mixture was then re-pyrolyzed at 800 °C using similar program as the Fe-ZIF-8-800 catalyst except another 70 °C hold for 1 h to remove the methanol.

## Data availability
The data supporting the findings of this study is available within the article and its Supplementary Information. All data used in this study are available in Google Drive (https://drive.google.com/drive/folders/107sONBFVChx7ETxNIzVAM-sAuP-TnvUB). All other relevant data is available from the corresponding authors upon reasonable request.

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

## Acknowledgements

Z. A., J. Li, T. W., Y. K., J. Liu, L. H., and Z. L. were supported by the National Key R&D Program of China (Grant No. 2022YFB3506200), National Natural Science Foundation of China (21702227, 22122113), Science Foundation of China University of Petroleum, Beijing (No. 2462014YJRC037). P. Y., S. C., and D. G. V. acknowledge support from the Catalysis Center for Energy Innovation, an Energy Frontier Research Center funded by the U.S. Department of Energy, Office of Science, Office of Basic Energy Sciences under Award number DE-SC0001004 for the computations. A. I. F. and S. X. acknowledge support by the U.S. National Science Foundation Grant CHE 2102299 (to A. I. F.). Dr. Jiang Li also thanks the support of the China Scholarship Council (CSC). Fe XAS measurements were performed at the beamline 1W1B of Beijing Synchrotron Radiation Facility (BSRF) in Beijing, China. C and N XAS experiments were performed at Beamlines MCD-A and MCD-B (Soochow Beamline for Energy Materials) at NSRL in Hefei, China. This research was supported in part from the Information Technologies (IT) resources at the University of Delaware, specifically the high-performance computing resources.

## Author contributions

Z.A., P.Y., and D.D. contributed equally to this work. J.Li, T.W., J.Liu, Z.L., and D.G.V. conceived the idea and designed the experiments. Z.A., P.Y., J.Li, and T.W. cowrite the manuscript. Z.A., T.W., and L.H. performed most of the experiments and characterizations. Z.A., D.D., Y.K., and R.L. carried out the EXAFS characterizations and analysis. P.Y., S.C., S.X., A.I.F., and Y.J. conducted the theoretical calculations. All the authors commented on the manuscript.

## Competing interests

The authors declare no competing interests.
