## [Peer Review File · Nature Communications]

Reviewers' comments:

Reviewer #1 (Remarks to the Author):

The authors developed a single-atom iron catalyst embedded in metal-organic framework, i.e., ZIF-8, for the catalytic transfer hydrogenation of biomass, i.e., furfural. They suggested the Fe(I)-pIN3 as an active site with unsaturated geometry was suggested though extensive theoretical and experimental studies, such as first principles simulations, spectroscopic calculations and experiments, and kinetic correlations. The synthesis of pyrrolic Fe(I)-pIN3 is of importance in the submitted study, which possesses sufficient flexibility for enhancing chemical reactivity. Furthermore, the catalytic activity of developed Fe-ZIF-8-800 catalyst was remarkable compared to previously reported catalysts. The computational studies reasonably explained the high catalytic activity of Fe-ZIF-8-800 catalyst using reaction energy variations and also electronic and geometric features based on crystal field theory. Therefore, I recommend the publication of submitted paper with following minor concerns, mainly regarding the explanation of Fe(I)-pIN3 formation.

(1) In Figure 1i, the author explained the higher binding energy of Zn in Fe-ZIF-8-800 than that in ZIF-8-800 with electron withdrawal from Zn to Fe. In the framework, Zn and Fe are separated with ligand species, and so the charge transfer seems to be improper. More reasonable explanations are required. Furthermore, the higher binding energy of Zn in ZIF-8 compared to that, i.e., ZnO, in MOF-5 was also used to suggest ZnN₄ species. Because ZnN₄ species has been studied in a lot of previous studies, the author should compare their data with previous one to support their claim of ZnN₄ formation.

(2) The formation of Fe(I)-pIN3 is a key finding of the submitted work for explaining the chemical reactivity of as-prepared Fe SAC embedded in ZIF-8. The main claim for the formation of Fe-pIN3 was provided based on EXAFS and XANES analyses because XPS of Fe was not detectable. In particular, the theoretical simulation of EXAFS and XANES spectra were used to support the formation of Fe(I)-pIN3, Fe(II)-pIN4, and Zn(II)-pIN4 which were also compared with those with pyridinic N. By the way, it is very difficult to find the difference of simulated spectra for Fe(I)-pIN3 (Fig. 2g) and Fe(II)-pIN4 (Fig S11a). The author should provide more explanations of this similarity.

(3) In computational results, the full structural model, i.e., the super cell, was not clearly provided in the paper. To clearly understand the computational approach, the authors need to provide it in SI. In addition, the atomic charges of Fe and Zn, i.e., Lewis acid sites, were also required to be provided during chemical reactions.

Reviewer #2 (Remarks to the Author):

An et al. presented the catalytic transfer hydrogenation of furfural and other substrates over single-atom Fe catalysts derived from Fe incorporated ZIF-8 with successive pyrolysis processes. The prepared catalysts were characterized by several analyses and showed high activity for catalytic reaction.

However, there are many claims about their results, therefore, I cannot recommend this study to be published in Nature Communication. Please check the raised issues as below:

- 1) I don't think defining the TOF value by 2min reaction result is reasonable (line 206).

- 2) It is really hard to believe that the authors made Fe-ZIF-8-800 catalysts with Fe loadings of 0.05, 0.10, 0.20, and 0.30. It is impossible to make such exact loading amounts by pyrolyzing Fe containing ZIF. The authors mentioned the synthesis procedure for different amounts of Fe catalysts by ex-situ impregnation in the ZIF-8-800 support. The Fe-ZIF-8-800 made by in situ synthesis and ex situ synthesis have totally different characteristics. Although the authors totally excluded the possibility of Fe coordination with 2-MeIM, there are many reports showing that Fe ion can be introduced in the ZIF-8 structure (Appl. Surf. Sci. 2022, 586, 152687, J. Hazard. Mater. 2021, 416, 126046, ACS Omega, 2021, 6, 31632, Adv. Funct. Mater. 31, 2009645) by in situ and ex situ synthesis.

- 3) Line 147-149, the authors explained that the electron withdrawal from Zn to Fe in Fe-ZIF-8-800 catalyst. Then Fe-ZIF-8-800 has Fe-Zn coordination? This is contradictory to the interpretation in the whole manuscript, especially EXAFS results.

- 4) Line 176, the Fe-Fe and Zn-Zn bonds are much shorter than 6A.

- 5) The authors should assign the acid and base sites from the TPD results (Figure S5) of Fe-ZIF-8-800, ZIF-8-800, and Fe-MOF-800, and correlate the catalytic results.

- 6) What is the origin of lattice oxygen detected by XPS (Figure S6) for Fe-ZIF-8-400 and Fe-ZIF-8-600?

7) The re-calcination process for the reused catalyst (Figure S21) is improper. Did the authors add Fe precursor on the spent catalyst? Then, it is not a regeneration process. Also, it's not a re-calcination but a re-pyrolysis process.

8) The higher yields of solvent-derived products than yields of furfuryl alcohol (entry 2, 4, 7, 12-15) means that the reaction pathway also includes dehydrogenation-hydrogenation?

9) How can the recycle results give only minor loss of activity even Fe leaching was 0.41 and 0.14 wt% after 1st and 2nd reaction (Table S15)?

10) How can the catalytic activity be recovered by re-activating the used catalyst with very low BET surface area (33 m²/g) compared to the fresh catalyst (403 m²/g) (Table S16)? It is also questionable that the surface area is increased after regeneration process (from 33 m²/g to 99 m²/g). If the surface area of support (ZIF-8-800) is reduced, the catalyst must have much larger amounts of Fe clusters or nanoparticles than the fresh sample. If the authors obtained moderate recyclability, it means that the suggested single atomic Fe site is not the major active site.

Reviewer #3 (Remarks to the Author):

This manuscript describes the transfer hydrogenation of furfural or other cyclic aldehyde with 2-propanol reductant and decomposed Fe-added ZIF-8 catalyst. The catalyst showed very high activity, and the authors speculate that the high activity is due to the tri-coordinated Fe(I) species. I agree with that this catalyst has high activity, although the selectivity and applicability (Table 1) are not so excellent. On the other hand, formation of tri-coordinated Fe(I) species was not solidly supported by the characterization, only XANES fitting. Fe(I) is a very rare species. If the formation is solidly confirmed, it attracts many readers with broad field. But, the evidence is not much. Rather, there are other results that oppose the formation of Fe(I), as listed below.

1. The coordination number of Fe-N in the catalyst was 3.70 +/- 0.56 and 4.56 +/- 0.58 for fresh and used catalysts, respectively. This means that tetra-coordinated species should be the main one. On the other hand, the XANES spectrum was fitted with tri-coordinated species, which means that the authors assume that the main species was tri-coordinated one. This is paradoxical.

2. Typical methods for valence state determination such as XPS (for Fe) and Mo:ssbauer spectroscopy were not tested. Experimental spin state determination is essential for Fe species.

3. The CN and bond length did not correspond to the valence state of 1. The well-known bond valence sum method gives 2-3 valence states from the CN and bond length data (Acta Crystallogr. B 1991, 47, 192).

4. The used catalyst was regenerated by addition of Fe(III) nitrate. The formation of Fe(I) in the used catalyst is not plausible.

Additional comments:

5. The authors stressed the importance of single-atom iron catalysts, even in the manuscript title. However, single-atom catalysis is common in this research field and not important; this catalyst is a solidified complex catalyst, and almost all complex catalysts are single-atom ones.

6. In the regeneration step, the authors stressed the small amount in the newly added Fe: "additional trace Fe precursor", "only 0.14 wt%". However, the added amount was about twice of the Fe amount in the fresh catalyst. It is not small.

7. The TOF value (2435 h^{-1}) was overestimated and too precise. This value was obtained only one run at very short reaction time (2 min), which can have large errors, including that derived by the reaction during the heating. According to Figure S13b, the conversion linearly increased until 15 min. The reaction rate should be calculated by the slope of the linear increase.

8. Table S10 should include simple homogeneous $\text{Fe}(\text{NO}_3)_3$ and $\text{Fe}(\text{acac})_3$ catalysts.

Below are our responses to the reviewers in a point-to-point fashion:

Reviewer #1 (Remarks to the Author):

1. The authors developed a single-atom iron catalyst embedded in metal-organic framework, i.e., ZIF-8, for the catalytic transfer hydrogenation of biomass, i.e., furfural. They suggested the Fe(I)-p1N₃ as an active site with unsaturated geometry was suggested though extensive theoretical and experimental studies, such as first principles simulations, spectroscopic calculations and experiments, and kinetic correlations. The synthesis of pyrrolic Fe(I)-p1N₃ is of importance in the submitted study, which possesses sufficient flexibility for enhancing chemical reactivity. Furthermore, the catalytic activity of developed Fe-ZIF-8-800 catalyst was remarkable compared to previously reported catalysts. The computational studies reasonably explained the high catalytic activity of Fe-ZIF-8-800 catalyst using reaction energy variations and also electronic and geometric features based on crystal field theory. Therefore, I recommend the publication of submitted paper with following minor concerns, mainly regarding the explanation of Fe(I)-p1N₃ formation.

Response: Thank you for your positive feedback and recommendation for publication. Upon further analysis, we confirmed that Fe species labeled as Fe(I)-p1N₃ should be Fe(II)-p1N₃ with an Fe oxidation state of +2. We have revised our manuscript to provide a more detailed explanation of the formation of pyrrolic Fe(II)-p1N₃. We believe that the revised manuscript addresses the concerns and provides a clearer understanding of the formation of this important coordination structure.

2. In Figure 1i, the author explained the higher binding energy of Zn in Fe-ZIF-8-800 than that in ZIF-8-800 with electron withdrawal from Zn to Fe. In the framework, Zn and Fe are separated with ligand species, and so the charge transfer seems to be improper. More reasonable explanations are required. Furthermore, the higher binding energy of Zn in ZIF-8 compared to that, i.e., ZnO, in MOF-5 was also used to suggest ZnN₄ species. Because ZnN₄ species has been studied in a lot of previous studies, the author should compare their data with previous one to support their claim of ZnN₄ formation.

Response: Thanks for your comment. Fu et al. (*Angew. Chem. Int. Ed.*, **2021**, *60*, 14005–14012) proposed that the Zn atoms can donate electrons to Cu atoms, resulting in a lower valence of Cu atoms in Cu/Zn-NC, which could favor O₂ adsorption and oxygen reduction activity. However, the direct Cu-Zn coordination structure was also absent in the Cu/Zn-NC catalysts according to the EXAFS fitting results. Charge transfer may also occur between the Zn-N_x and Fe-N_x species rather than directly through Fe-Zn coordination. We have revised the corresponding discussion in the 3rd paragraph of the catalyst synthesis and characterization section in the revised manuscript. It reads: “The slightly higher binding energy of Zn in Fe-ZIF-8-800 than ZIF-8-800 (Figure 1i) is attributed to electron withdrawal from Zn to Fe even though there are no direct Zn-Fe bonds as shown in EXAFS spectra (Figure 2c). That is possible as charge transfer can occur through metal-N bonds, similarly to the N-mediated charge transfer in the Cu/Zn-NC catalyst where direct Cu-Zn coordination is absent according to the EXAFS results [29].”

Furthermore, we agree that comparing our data with previous studies on ZnN₄ species is crucial to support our claim of its formation. Therefore, we have compared our results with

previous studies in more detail to substantiate the existence of ZnN_4 species in our catalysts. Zeng et al. (*Small Struct.*, **2022**, 3, 2100225) investigated the atomic states of metal in COF@MOF_{800} and $\text{COF@MOF}_{800}\text{-Fe}$ catalysts via XAFS. The main peak position of Zn in the R-space was 1.53 Å for both catalysts. Fitting the EXAFS of COF@MOF_{800} reveals that Zn atoms were coordinated as Zn-N with a coordination number of 4.1 and an average bond distance of 1.98 Å. $\text{COF@MOF}_{800}\text{-Fe}$ showed similar Zn XAFS spectra to COF@MOF_{800} , confirming the ZnN_4 structure in carbon. Wang et al. (*Angew. Chem. Int. Ed.*, **2020**, 59, 22408–22413) investigated the local structure of Zn-N-C via Zn K-edge XANES and EXAFS. Fourier transformation of EXAFS analysis shows that Zn-N-C possesses a peak around 2.0 Å corresponding to Zn-N bonds. The fitting results show that the coordination number of Zn sites in Zn-N-C is about 3.9. In our original manuscript, Zn K-edge XANES and EXAFS spectra were also collected to investigate the local structure of ZIF-8-derived catalysts. The coordination number of the Zn site in the ZIF-8-800 catalyst and fresh and used Fe-ZIF-8-800 catalyst are 4.1, 3.9, and 4.1, respectively, and the bond distances are 1.97 Å, 1.98 Å, and 1.98 Å, respectively, which are highly consistent with previous reports and strongly support the formation of ZnN_4 species in ZIF-8-derived catalysts.

3. The formation of Fe(I)-pIN₃ is a key finding of the submitted work for explaining the chemical reactivity of as-prepared Fe SAC embedded in ZIF-8. The main claim for the formation of Fe-pIN₃ was provided based on EXAFS and XANES analyses because XPS of Fe was not detectable. In particular, the theoretical simulation of EXAFS and XANES spectra were used to support the formation of Fe(I)-pIN₃, Fe(II)-pIN₄, and Zn(II)-pIN₄ which were also compared with those with pyridinic N. By the way, it is very difficult to find the difference of simulated spectra for Fe(I)-pIN₃ (Fig. 2g) and Fe(II)-pIN₄ (Fig S11a). The author should provide more explanations of this similarity.

Response: We thank the reviewer for this comment. We concede that the simulated XANES spectra of the species we labeled Fe(I)-pIN₃ and Fe(II)-pIN₄ are similar. The particular labeling was based on the defective porphyrin-based structure and implied different formal oxidation states. As stated in the reply to the first comment, upon further analysis prompted by the reviewer's comment, we believe that the two species are in the same oxidation state +2, which should explain the similar XANES spectra. Specifically, the DFT calculated magnetic moments of the planar, T-shaped Fe center in the clean surface and the tetrahedral Fe center in the intermediate depIPA_H_FF* (Fe coordinated to deprotonated IPA and an FF molecule) are 3.28 μ_B and 3.34 μ_B , respectively. These values suggest high-spin $3d^6 \text{Fe}^{2+}$ with 4 unpaired electrons or high-spin $3d^7 \text{Fe}^{1+}$ with 3 unpaired electrons. However, the $3d^7 \text{Fe}^{1+}$ possibility should be discarded because the square planar Fe center in depIPA_H_FF* shows a magnetic moment of 2.07 μ_B , corresponding to only 2 unpaired electrons, namely the low-spin $3d^6 \text{Fe}^{2+}$ configuration. Taken together, we conclude that the two species, originally labeled Fe(I)-pIN₃ and Fe(II)-pIN₄, are in the same oxidation state +2.

We have added text to explain this important aspect in the 6th paragraph of the reaction mechanism section on page 18 in the revised manuscript. It reads: "To gain insights into the higher activity of Fe(II)-pIN₃ relative to the Fe-site labeled Fe(II)-pIN₄, we investigated the electronic configurations of the two Fe species. Upon careful analysis, we believe the two Fe species are in the same oxidation state +2. The DFT magnetic moment of the square planar Fe center in the clean surface of Fe(II)-pIN₄ is 1.89, indicating a low-spin $3d^6 \text{Fe}^{2+}$ configuration. For the Fe-pIN₃ site,

the planar, T-shaped Fe center in the clean surface and the tetrahedral Fe center in the intermediate depIPA_H_FF* (Fe coordinated to deprotonated IPA and an FF molecule) have magnetic moments of 3.28 μ_B and 3.34 μ_B , respectively. These values suggest high-spin 3d⁶ Fe²⁺ with 4 unpaired electrons or high-spin 3d⁷ Fe¹⁺ with 3 unpaired electrons. However, the 3d⁷ Fe¹⁺ possibility should be discarded because the square planar Fe center in depIPA_H_FF* shows a magnetic moment of 2.07 μ_B , corresponding to only 2 unpaired electrons, namely the low-spin 3d⁶ Fe²⁺ configuration. Taken together, we conclude that the oxidation state of the Fe center in the Fe-pIN₃ site is +2, which is the same as that in the Fe-pIN₄ site. Nevertheless, the two Fe species exhibit different local structures.”

4. In computational results, the full structural model, i.e., the super cell, was not clearly provided in the paper. To clearly understand the computational approach, the authors need to provide it in SI. In addition, the atomic charges of Fe and Zn, i.e., Lewis acid sites, were also required to be provided during chemical reactions.

Response: We apologize for the oversight. The requested simulation cell information is now provided in the DFT Calculations and Microkinetic Modeling section on page 28 in the revised manuscript. The added text reads: “The supercells of Fe(II)-pIN₃, Fe(II)-pIN₄, and Zn(II)-pIN₄ were built based on 3×7 rectangular unit cells of graphene and have sizes of 12.8×17.2 Å.”

The structures of all intermediates were already provided as supporting information (Figure S17-S20 in the revised supporting information).

Additionally, the requested Bader charges of Fe and Zn are now provided in the supplementary Table S18 and are also shown as Table 1 below, and discussed in the Reaction Mechanism section on page 15 in the revised manuscript. The added text reads: “The Bader charge analysis (Table S18) confirms that the Fe and Zn atoms carry a positive partial charge and thus can be viewed as Lewis acidic centers.”

Table 1. Bader charges of Fe and Zn sites for each state in Fe(II)-pIN₃, Fe(II)-pIN₄, Zn(II)-pIN₄

Fe(II)-pIN ₃	
State	Bader Charge of Fe
*(clean surface)	1.08
IPA *	1.20
TS1	1.22
depIPA_H*	1.18
depIPA_H_FF*	1.35
TS2	1.32
ACE_H_FFH*	1.31
H_FFH*	1.12
TS3	1.18
FA *	1.16
Fe(II)-pIN ₄	
State	Bader Charge of Fe
*(clean surface)	1.16
FF*	1.23
IPA_FF*	1.16

TS(P1)	1.27
ACE_FA*	1.24
FA*	1.23
IPA*	1.16
FF_IPA*	1.18
TS(P2)	1.25
FA_ACE*	1.25
ACE*	1.24
Zn(II)-pIN ₄	
State	Bader Charge of Zn
*(clean surface)	1.16
FF*	1.24
IPA_FF*	1.23
TS(P1)	1.24
ACE_FA*	1.23
FA*	1.24
IPA*	1.23
FF_IPA*	1.25
TS(P2)	1.25
FA_ACE*	1.24
ACE*	1.25

Reviewer #2 (Remarks to the Author):

1. An et al. presented the catalytic transfer hydrogenation of furfural and other substrates over single-atom Fe catalysts derived from Fe incorporated ZIF-8 with successive pyrolysis processes. The prepared catalysts were characterized by several analyses and showed high activity for catalytic reaction.

However, there are many claims about their results, therefore, I cannot recommend this study to be published in Nature Communication. Please check the raised issues as below:

Response: We appreciate your comments and have taken note of your concerns regarding the claims made in our manuscript. We understand the importance of providing a clear and concise description of our results and have carefully addressed each issue raised in your review.

2. I don't think defining the TOF value by 2min reaction result is reasonable (line 206).

Response: Thanks for your comment. In our original manuscript, we calculated the TOF value at 2 min because the FF conversion exceeds 20% over the highly active Fe-ZIF-8-800 at 120 °C quickly (*e.g.*, in 5 min). The TOF should be calculated at a conversion of <20% in the kinetic regime (*J. Catal.*, **1966**, 6, 92–99; *Adv. Catal.*, **1969**, 20, 153–166). We have added results and discussion about the TOF at a lower reaction temperature (80 °C) with longer reaction time to avoid the short reaction time issue.

As shown in Figure 1 below (Figure S15 in the revised supporting information), the FF conversion correlates well with reaction time at times shorter than 60 min at 80 °C and 120 °C. The TOF value is almost *vs.* time even at longer times. Indeed, TOF values at various FF

conversions and times at 80 °C and 120 °C over Fe-ZIF-8-800 are shown in Table 2 below (Table S17 in the revised supporting information).

$$\text{TOF} = \frac{\text{molar amount of FF consumed at low conversions}}{\text{molar amount of Fe in catalysts} \times \text{reaction time}} \times 100\%$$

Figure 1. Time course of FF conversion over Fe-ZIF-8-800 at 80 °C and 120 °C.

Table 2. Catalytic transfer hydrogenation of FF at various reaction conditions.^a

Entry	T [°C]	t [min]	Conv. [%]	Yield [%]	TOF [h ⁻¹]
1	80	15	14.5	6.1	443.7
2	80	30	31.1	21.2	475.8
3	80	60	63.3	50.5	484.2
4	120	5	20.5	14.1	1881.9
5	120	10	33.3	23.3	1528.5
6	120	15	49.1	39.9	1502.5

^aReaction conditions: 0.5 mmol FF, 3 mL isopropanol solvent, 50 mg catalyst (0.13 mol% Fe). TOF was calculated in h⁻¹ as in the original manuscript.

Although the reaction temperature for the TOF calculation of our Fe catalyst was as low as 80 °C, the TOF is still two to three orders of magnitude higher than other Fe-based catalysts (Table S16 in the revised supporting information) calculated in the range of 160–200 °C. It is more accurate to estimate the TOF value of Fe-ZIF-8-800 from these longer times as suggested by the reviewer (Figure 1 and Table 2 here or Figure S15 and Table S17 in the revised supporting information). We have revised the corresponding results and discussion on page 13, lines 1–10 in the revised manuscript.

3. It is really hard to believe that the authors made Fe-ZIF-8-800 catalysts with Fe loadings of 0.05, 0.10, 0.20, and 0.30. It is impossible to make such exact loading amounts by pyrolyzing Fe containing ZIF. The authors mentioned the synthesis procedure for different amounts of Fe catalysts by ex-situ impregnation in the ZIF-8-800 support. The Fe-ZIF-8-800 made by in situ synthesis and ex situ synthesis have totally different characteristics. Although the authors totally excluded the possibility of Fe coordination with 2-MeIM, there are many reports showing that Fe ion can be introduced in the ZIF-8 structure (*Appl. Surf. Sci.*, **2022**, 586, 152687; *J. Hazard. Mater.*, **2021**, 416, 126046; *ACS Omega*, **2021**, 6, 31632; *Adv. Funct. Mater.*, 31, 2009645) by in situ and ex situ synthesis.

Response: Thanks for your comment. This is a good point that deserves explanation. We initially tried one-step method to synthesize the Fe-ZIF-8-800 catalyst, and the catalyst exhibited excellent performance for the catalytic transfer hydrogenation. However, ICP-AES showed strangely Fe loading of <0.1 wt% even though 175 mg Fe salts were added (actually we originally attempted synthesize a Fe catalyst with a loading about 5% from many Fe precursors). The Fe loading was confirmed several times. As Fe is impossible to construct in the ZIF-series MOFs (*Science*, **2008**, 319, 939–943), we proposed that Fe is adsorbed in the ZIF-8 crystals. We agree that Fe can be introduced to the final catalysts in single atomic state through confining into the cages of ZIF-8, but Fe atoms in the framework of ZIF are impossible.

Given the adsorption amounts of Fe are low and uncontrollable, we fabricated the 0.1%Fe-ZIF-8-800 catalyst via two-step route. The weight of ZIF-8 precursor decreased from 660.0 to 324.4 mg as ZIF-8-800 catalyst after the first pyrolysis at 800 °C. The mass loss ratio is 50.8%. In the second pyrolysis, the mass of 0.1%Fe-ZIF-8-800 catalyst is 339.4 mg from 341.3 mg of ZIF-8-800 and Fe(NO₃)₃, and the mass loss ratio is only 0.6%. Therefore, the main mass loss occurs in the first step pyrolysis. Thus, the theoretical Fe loading is nearly the same as the actual Fe content using the two-step route. ICP-AES was used to determine the actual Fe loading in some Fe-ZIF catalysts (shown in Table 3 below and Table S2 in the revised supporting information).

Table 3. Contents of Fe in various two-step Fe-ZIF catalysts.

Entry	Catalyst	Fe Content [%]
1	0.1%Fe-ZIF-8-800	0.08
2	0.3% Fe-ZIF-8-800	0.29

Thus, two-step method was applied to precisely tune the Fe loading and confirm that Fe species are the real active centers. In the revised manuscript, we compare in detail catalysts prepared from one-step and two-step methods, including the same single-atomic Fe state and comparable performance to confirm they are the same catalysts even though prepared via different methods. The similar performance also confirmed our hypothesis about the saturated adsorption of Fe onto ZIF-8 in one-step synthesis.

Firstly, we investigated the CTH performance of the two catalysts at 80 °C and 120 °C, respectively, as shown in Table 4 (Table S3 in the revised supporting information). The Fe-ZIF-8-800 catalysts synthesized via in-situ (one-step) and ex-situ (two-step) impregnation exhibit similar high catalytic activity (FF conversion, FA yield, and TOF) under identical reaction conditions.

Table 4. CTH performance of FF over one- and two-step Fe-ZIF-8-800 catalysts^a.

Entry	Catalyst	T [°C]	T [min]	Conv. [%]	Yield [%]	TOF [h ⁻¹]
1	Fe-ZIF-8-800 ^b	80	30	24.0	17.6	367.2
2		120	360	99.6	96.5	1881.9 ^c
3	Fe-ZIF-8-800 ^d	80	30	23.8	13.8	320.7
4		120	360	96.7	93.8	1843.1 ^c

^aReaction conditions: 0.5 mmol FF, 3 mL isopropanol solvent, 50 mg catalyst. ^bPrepared via the one-step route. ^cThe TOF was calculated based on the FF conversion at 5 min. ^dPrepared via the two-step route.

Secondly, insights into the state of Fe in one-step and two-step Fe-ZIF-8-800 catalysts were gained via X-ray absorption near-edge structure (XANES) and Fourier transforms (FTs) of EXAFS spectra during this manuscript revision. In these two Fe-ZIF-8-800 catalysts, Fe atoms are both oxidized (Figure 2a in this document, also shown as Figure S1a in the revised supporting information), and the lengths of Fe-N bonds are both *ca.* 1.5 Å (Figure 2b in this letter, also shown as Figure S1b in the revised supporting information). Wavelet transform spectra further confirm that the Fe atom is well dispersed without aggregation in both catalysts (Figure 2e in this letter, also shown as Figure S1e in the revised supporting information). In addition, the Fe-N coordination numbers in two Fe-ZIF-8-800 catalysts are both *ca.* 3.5 (Table 5 in this letter, also shown as Table S4 in the revised supporting information), suggesting the same single-atomic Fe state in the Fe-ZIF-8-800 catalysts prepared by one- and two-step routes. The Fe-N coordination numbers obtained this time is slightly less than that in previous manuscript due to different radiation source or different batches of catalysts.

Figure 2. (a) Fe K-edge normalized X-ray absorption near-edge structure (XANES) spectra. (b) Fourier transform of k^2 -weighted Fe K-edge EXAFS spectra. EXAFS fitting of the one-step (c) and two-step (d) Fe-ZIF-8-800 catalysts at Fe K-edge. (e) Wavelet transform of Fe K-edge

EXAFS for one-step and two-step Fe-ZIF-8-800 catalysts.

Table 5. Fitting results of Fe K-edge EXAFS data of one- and two-step Fe-ZIF-8-800 catalysts.

Sample	Scattering Path	CN	R(Å)
Fe-ZIF-8-800 ^a	Fe-N	3.5 ± 0.7	1.97 ± 0.04
Fe-ZIF-8-800 ^b	Fe-N	3.6 ± 0.5	1.97 ± 0.05

^aOne-step Fe-ZIF-8-800; ^btwo-step Fe-ZIF-8-800; CN, the coordination numbers; R, the bonding distance.

Taken together, we have confirmed that the catalysts prepared from in-situ and ex-situ methods with the same Fe loading exhibit almost the same performance and possess the same Fe states. The Fe loading is highly controllable via the ex-situ method, and the linear correlation between the Fe loading and specific reaction rates clearly shows that Fe species are the active sites in the prepared Fe catalysts.

4. Line 147-149, the authors explained that the electron withdrawal from Zn to Fe in Fe-ZIF-8-800 catalyst. Then Fe-ZIF-8-800 has Fe-Zn coordination? This is contradictory to the interpretation in the whole manuscript, especially EXAFS results.

Response: Thanks for your comment. Reviewer 1 raised the same concern as you (No. 2), and we have addressed this point above. Please refer our response there.

5. Line 176, the Fe-Fe and Zn-Zn bonds are much shorter than 6 Å.

Response: Thanks for your comment. In the original manuscript, we stated: “Wavelet transform spectra further confirm that Fe and Zn atoms are well dispersed without aggregation (Figure 2i), whereas Fe-Fe and Zn-Zn scatterings are observed at about 6 Å in the Fe-MOF-5-800 catalyst (Figure S9g).”

Firstly, we apologize for the description in the original manuscript. The “6 Å” is of course not reasonable (should read below 6 Å). Instead, we should have written that the Fe-Fe and Zn-Zn scatterings are observed in the high *k* value range in Fe and Zn K-space spectra. Secondly, the R-space was obtained by Fourier transform in the K-space, which can represent the bond distance. According to Figure S10g in the revised supporting information, the Fe-Fe bond is *ca.* 2.5 Å, and the Zn-Zn bond is *ca.* 2.75 Å, which are indeed shorter than 6 Å. We have revised the manuscript accordingly. See page 11, lines 12–15 in the revised manuscript for details.

6. The authors should assign the acid and base sites from the TPD results (Figure S5) of Fe-ZIF-8-800, ZIF-8-800, and Fe-MOF-800, and correlate the catalytic results.

Response: Thanks for your advice. We have correlated the catalytic performance with TPD results and added more discussions in the revised manuscript. First, the CTH performance follows the order of Fe-ZIF-8-800, ZIF-8-800 and Fe-MOF-5-800 as shown in Table S13 in the revised supporting information. There are no acidic or basic sites in Fe-MOF-5-800 catalysts, consistent with its poor CTH activity. The total acidic and basic sites of ZIF-8-800 catalyst are 69.2 and 175.7, respectively. The Fe-ZIF-8-800 catalyst possesses more acidic sites (139.8) and basic sites (292.9), giving the best CTH performance. In the revised manuscript, we added the following: “The Fe-MOF-5-800 and ZIF-8-800 catalysts give lower FF conversion and FA selectivity and

TOF of *ca.* 0.054 h⁻¹, indicating that the MOF-5 precursor and the absence of Fe lead to inferior catalytic activity, consistent with their poor acid-base properties (Figure S14).” In addition, in the second paragraph in the reaction mechanism (page 15), the Bader charge analysis (Table S17) confirms that the Fe and Zn atoms carry positive partial charge and thus can be viewed as Lewis acidic centers.

7. What is the origin of lattice oxygen detected by XPS (Figure S6) for Fe-ZIF-8-400 and Fe-ZIF-8-600?

Response: Thanks for your comment. We have characterized the Fe-ZIF-8-400 and Fe-ZIF-8-600 catalysts again via XPS, and the peaks that previously attributed to lattice oxygen were still observed. After reviewing literature, the peak should be assigned as carbonate formed from pyrolysis of the ZIF-8 precursor in an inert atmosphere. Lin et al. (*J. Phys. Chem. C*, **2016**, *120*, 14015–14026) assessed the rate of weight change in ZIF-8 crystals under inert atmosphere at elevating temperatures to develop ZIF-8 thermal decomposition kinetics and deduced the resulting carbonate structure after decomposition. The peak at *ca.* 530.9 eV in the O 1s XPS spectra of ZIF-8 samples was attributed to carbonate by Figueiredo et al. (*Nanomaterials*, **2019**, *9*, 1369). In addition, we have also re-characterized Fe-ZIF-8-800 catalyst via XPS as shown in Figure 3 in this letter (Figure S6 in the revised supporting information). We have revised the corresponding peak assignment in the revised supporting information.

Figure 3. O 1s XPS spectra of various MOF-derived catalysts.

8. The re-calcination process for the reused catalyst (Figure S21) is improper. Did the authors add Fe precursor on the spent catalyst? Then, it is not a regeneration process. Also, it's not a re-calcination but a re-pyrolysis process.

Response: Thanks for this point. We apologize for the confusion regarding the catalyst regeneration process. We agree that the process described in Figure S24 in the revised supporting information is not a re-calcination but rather re-pyrolysis. Additionally, we did add Fe precursor to the spent catalyst during the re-pyrolysis. We have revised the manuscript to reflect these corrections and clarified the regeneration process.

Furthermore, in the revised manuscript, we have improved the regeneration step of Fe-ZIF-8-800 catalyst according to the comments from Reviewer 3 (comment 6), in which the amount of Fe precursor to regenerate the activity is significantly decreased from 0.14 wt% to 0.01 wt%, fourteen times less. As shown in Figure S24a in the revised supporting information (also shown in Figure 4 in this letter), the CTH activity was completely recovered compared with the fresh Fe-ZIF-8-800, confirming that trace additional Fe species can completely recover the activity of Fe-ZIF-8-800.

Figure 4. (a) Recyclability tests of Fe-ZIF-8-800 catalyst. Reaction conditions: 0.5 mmol FF, 3 mL isopropanol solvent, 120 °C, 15 min, 50 mg Fe-ZIF-8-800 catalyst for the first run, and then reused for 4 runs. To recover the activity of Fe-ZIF-8-800 catalyst, re-pyrolysis of the reused catalyst is proceeded by adding 0.033 wt% $\text{Fe}(\text{NO}_3)_3$ (the Fe dosage is only 0.01 wt%).

9. The higher yields of solvent-derived products than yields of furfuryl alcohol (entry 2, 4, 7, 12–15) means that the reaction pathway also includes dehydrogenation-hydrogenation?

Response: Thanks for your comment. Dehydrogenation of alcoholic solvents is possible to occur over Fe catalysts. In Figure 3b of the original manuscript, for the calculation of generation rate of acetone, no furfural was added, *i.e.*, only isopropanol was used as the substrate. The linear correlation between Fe loading and specific reaction rate of acetone clearly indicated that the Fe-ZIF-8-800 catalyst has independent dehydrogenation ability. However, the specific rate of

acetone is *ca.* 5 times lower than that of furfural or furfuryl alcohol, indicating transfer hydrogenation is dominant in our work than the dehydrogenation-hydrogenation route.

10. How can the recycle results give only minor loss of activity even Fe leaching was 0.41 and 0.14 wt% after 1st and 2nd reaction (Table S15)?

Response: Thanks for your comment. We apologize for the confusion in the description of Fe leaching during the recyclability experiments of the Fe-ZIF-8-800 catalyst. In fact, the leaching of Fe was only 0.000299 wt% ($0.073 \text{ wt}\% \times 0.41 \text{ wt}\%$) after the first cycle, and the leaching of Fe was 0.000102 wt% ($0.073 \text{ wt}\% \times 0.14 \text{ wt}\%$) after the second cycle. Therefore, the actual Fe leaching was very low. We have corrected corresponding results in Table S20 in the revised supporting information.

11. How can the catalytic activity be recovered by re-activating the used catalyst with very low BET surface area ($33 \text{ m}^2/\text{g}$) compared to the fresh catalyst ($403 \text{ m}^2/\text{g}$) (Table S16)? It is also questionable that the surface area is increased after regeneration process (from $33 \text{ m}^2/\text{g}$ to $99 \text{ m}^2/\text{g}$). If the surface area of support (ZIF-8-800) is reduced, the catalyst must have much larger amounts of Fe clusters or nanoparticles than the fresh sample. If the authors obtained moderate recyclability, it means that the suggested single atomic Fe site is not the major active site.

Response: Thanks for your comment. We had confirmed the single atomic state of Fe in the re-used Fe-ZIF-8-800 catalyst (Figure S22 in the original supporting information). Thus, the decrease of BET surface area suggests the pore structure of the support (ZIF-8-800) greatly changed, due possibly to blocking by residues or oligomers leading to Fe being less exposed and inferior catalytic performance during recycling. To re-activate the Fe catalyst, trace Fe salts are added before re-pyrolysis. In this case, the BET surface area of the catalyst increased from $33 \text{ m}^2/\text{g}$ to $99 \text{ m}^2/\text{g}$, and the catalyst activity is completely recovered. In the revised manuscript, the amount of additional Fe precursors is further decreased from 0.14 wt% to 0.01 wt%, and the BET surface area of the re-pyrolyzed Fe-ZIF-8-800 is $178 \text{ m}^2/\text{g}$ as shown in Table S21 in the revised supporting information. The re-pyrolysis process with trace additional Fe precursors increased the proportion of unsaturated Fe-N₃ coordination and improved the pore structure, thereby improving the catalytic activity. Besides this, we have previously confirmed that single-atom Fe species were the active sites over Fe-ZIF-8-800 via EXAFS (Figure 2 in the revised manuscript) and the linear correlation between Fe loading and specific reaction rate (Figure 3b in the revised manuscript).

Reviewer #3 (Remarks to the Author):

1. This manuscript describes the transfer hydrogenation of furfural or other cyclic aldehyde with 2-propanol reductant and decomposed Fe-added ZIF-8 catalyst. The catalyst showed very high activity, and the authors speculate that the high activity is due to the tri-coordinated Fe(I) species. I agree with that this catalyst has high activity, although the selectivity and applicability (Table 1) are not so excellent. On the other hand, formation of tri-coordinated Fe(I) species was not solidly supported by the characterization, only XANES fitting. Fe(I) is a very rare species. If the formation is solidly confirmed, it attracts many readers with broad field. But, the evidence is not much. Rather, there are other results that oppose the formation of Fe(I), as listed below.

Response: Thanks for finding the catalyst activity high and the paper attractive to many readers in

the broad field. We appreciate bringing up the apparent inconsistency on the oxidation state of Fe. This is the same point as comment 3 of Reviewer 1. Please see the reply above.

2. The coordination number of Fe-N in the catalyst was 3.70 +/- 0.56 and 4.56 +/- 0.58 for fresh and used catalysts, respectively. This means that tetra-coordinated species should be the main one. On the other hand, the XANES spectrum was fitted with tri-coordinated species, which means that the authors assume that the main species was tri-coordinated one. This is paradoxical.

Response: We thank the reviewer for this comment. We wish to point out that we simulated XANES Fe-N_x spectra for several species and coordination numbers and not just for the Fe-p1N₃ species. Using the simulated XANES spectra, we demonstrated that both Fe-p1N₃ and Fe-p1N₄ are good representations of the Fe-N_x active sites. We agree that the coordination numbers inferred from the experimental measurements suggest that the Fe-p1N₄ site could be dominant in concentration. However, according to the DFT calculations and microkinetic simulations, the TOF of Fe-p1N₄ is nearly 6 orders of magnitude lower than that of the Fe-p1N₃ site (Table S19 in the revised supporting information). Therefore, we believe that our simulations show that the minority Fe-p1N₃ sites are the active sites.

3. Typical methods for valence state determination such as XPS (for Fe) and Mossbauer spectroscopy were not tested. Experimental spin state determination is essential for Fe species.

Response: Thanks for your comment. In the original manuscript, we have stated that: “X-ray absorption near-edge structure (XANES) and Fourier transforms (FTs) of EXAFS spectra provide insights into the state of Fe even at low fractions as aberration-corrected transmission electron microscopy (Figure S8) cannot distinguish Fe and Zn atoms due to approximate atomic number [31].” The extremely low loading of Fe leads to challenges in exploring the state of Fe by traditional methods such as XPS. We have now provided the Fe 2p XPS of Fe-ZIF-8-800 catalyst as shown in Figure 5 in this letter (figure S9 in the revised supporting information). No valuable or significant signals are observed in Fe 2p XPS of Fe-ZIF-8-800 catalyst, indicating that XPS analysis is not suitable for the Fe-ZIF-8-800 catalyst with ultra-low Fe loading. Furthermore, we agree that the experimental spin state determination is essential for Fe species. To address this point, we have inquired with multiple characterization facilities for Fe Mossbauer characterization (the applicable Fe content is generally >5 wt%) of the Fe-ZIF-8-800 catalyst. However, they were unable to provide Fe Mossbauer spectra with good signals, similar to the Fe 2p XPS, as shown in Figure 5 in this letter, due to the extremely low Fe content (0.073 wt%) even using a long testing cycle (10–12 days) and a high testing cost (ca. 20,000 RMB).

Figure 5. Fe 2p XPS spectra of Fe-ZIF-8-800 catalyst.

4. The CN and bond length did not correspond to the valence state of 1. The well-known bond valence sum method gives 2–3 valence states from the CN and bond length data (Acta Crystallogr. B, 1991, 47, 192).

Response: We appreciate the comment. We apologize for the oversight in the original version. See comment 1 and our response to comment 3 of Reviewer 1.

5. The used catalyst was regenerated by addition of Fe(III) nitrate. The formation of Fe(I) in the used catalyst is not plausible.

Response: Thanks for your comment. We first confirmed that Fe(I) species are not present in our catalyst, and the Fe species should be Fe(II)-pIN₃ with an Fe oxidation state of +2. Secondly, in the regeneration of used catalysts, we added trace amount of additional Fe(III) nitrate (additional Fe dosage is only 0.01 wt%) followed by the re-pyrolysis to recover the activity. During the pyrolysis, Fe(III) can be reduced to Fe(II) due to the reducibility of carbon-based supports. We have made a clearer description about this.

Additional comments:

6. The authors stressed the importance of single-atom iron catalysts, even in the manuscript title. However, single-atom catalysis is common in this research field and not important; this catalyst is a solidified complex catalyst, and almost all complex catalysts are single-atom ones.

Response: Thanks for your comment. It is true that the fabrication of single-atom (SA) catalysts draws a lot of ideas from homogeneous complex catalysts, and the active center of many SA catalysts is similar to complex catalysts, especially when catalyzing organic synthesis (*Angew.*

Chem. Int. Ed., e202219306). Recently, there are many excellent work and reviews on SA catalysts, so that the importance of SA catalysis is unquestionable due to the clear coordination environment of the active center to establish structure-property relationships and elucidate the reaction mechanism. Moreover, the application of SA catalysts in biomass conversions, especially Fe-based SA catalysts are still rare, *e.g.*, *Chem. Soc. Rev.*, **2020**, *49*, 3764–3782). We felt that the title aligns well with how the SA term is used in the literature.

7. In the regeneration step, the authors stressed the small amount in the newly added Fe: "additional trace Fe precursor", "only 0.14 wt%". However, the added amount was about twice of the Fe amount in the fresh catalyst. It is not small.

Response: Thanks for your comment. We have significantly decreased the amount of additional Fe to 0.01 wt%, and the activity is still successfully recovered. Please see our reply to comment 8 of Reviewer 2 as same concerns are raised by you.

8. The TOF value (2435 h^{-1}) was overestimated and too precise. This value was obtained only one run at very short reaction time (2 min), which can have large errors, including that derived by the reaction during the heating. According to Figure S13b, the conversion linearly increased until 15 min. The reaction rate should be calculated by the slope of the linear increase.

Response: Thanks for this point. See our reply to Reviewer 2, comment 2 for the response to same concern.

9. Table S10 should include simple homogeneous $\text{Fe}(\text{NO}_3)_3$ and $\text{Fe}(\text{acac})_3$ catalysts.

Response: Thanks for your advice. We have investigated the CTH performance of simple homogeneous $\text{Fe}(\text{NO}_3)_3$ and $\text{Fe}(\text{acac})_3$ catalysts. Corresponding results (Table 6 in this letter) have been added in Table S13 in the revised supporting information.

Table 6. CTH performance of FF over simple homogeneous $\text{Fe}(\text{NO}_3)_3$ and $\text{Fe}(\text{acac})_3$ catalysts.^a

Entry	Catalyst	T [°C]	t [h]	Conv. [%]	Yield [%]	Sel. [%]	Yield of solvent-derived products	
								1	$\text{Fe}(\text{NO}_3)_3$	120	6	4.9	0.6	12.2	5.9	0
2	$\text{Fe}(\text{acac})_3$	120	6	11.8	3.0	25.4	3.6	<0.1

^aThe dosage of Fe is the same as Fe-ZIF-8-800 catalyst.

REVIEWERS' COMMENTS

Reviewer #3 (Remarks to the Author):

In my previous review, I wrote that "Fe(I) is a very rare species. If the formation is solidly confirmed, it attracts many readers with broad field." During the revision, the authors changed the proposed electronic state of catalyst, from Fe(I) to Fe(II). The assignment is reasonable...but the catalyst is not attractive in structural aspect. The value of this study is now simply evaluated by the performance. In view of performance, surely the activity was high, but the selectivity and applicability (Table 1) are not so excellent. Now, this work does not attract broad interest. Submission to specialized journal is recommended.

Below are our responses to the reviewer:

Reviewer #3 (Remarks to the Author):

1. In my previous review, I wrote that "Fe(I) is a very rare species. If the formation is solidly confirmed, it attracts many readers with broad field." During the revision, the authors changed the proposed electronic state of catalyst, from Fe(I) to Fe(II). The assignment is reasonable...but the catalyst is not attractive in structural aspect. The value of this study is now simply evaluated by the performance. In view of performance, surely the activity was high, but the selectivity and applicability (Table 1) are not so excellent. Now, this work does not attract broad interest. Submission to specialized journal is recommended.

Response: Thanks for your comment. We appreciate your previous inquiry into the Fe(I) species. Concerning your comment, we carefully reviewed relevant results and discussions, and a reasonable pyrrolic Fe(II) site is further proposed.

Although the Fe(II) species are not as rare as Fe(I), the precise identification of the active sites in this catalyst is attractive to readers. The Fe(II)-N₄ coordination has generally been considered as the active site for the adsorption and reduction of molecular oxygen in electrocatalysis such as oxygen reduction reaction (ORR) (*Nat. Mater.*, **2015**, *14*, 937–942; *Angew. Chem. Int. Ed.*, **2017**, *56*, 6937–6941; *Nat. Catal.*, **2018**, *1*, 63–72; *ChemElectroChem*, **2019**, *6*, 304–315). Our DFT, microkinetic modeling, kinetic experiments, and XANES simulations provide a systematic methodology to determine the active site. We propose for the first time the Fe(II)-pIN₃ site as the active one; it allows the co-adsorption of furoxy species and hydroxyl groups to support the typical MPV mechanism, as the Fe atom is anchored to a defect site and N participates in H transfer. In contrast, Fe(II)-N₄ cannot simultaneously coordinate furfural and isopropanol due to steric hindrance stemming from the rigid geometry of the metal-N₄ site.

Furthermore, we appreciate your acknowledgment of the outstanding activity of the prepared single-atom Fe-based catalyst, which undoubtedly underscores the value of this study. As summarized in Tables S15 and S16, the CTH activity of the Fe-ZIF-8-800 catalyst is superior to that of the reported Fe-based catalysts and comparable to state-of-the-art catalysts. Regarding the selectivity and applicability in Table 1, we also proposed a unique trait of our catalyst whereby the chemistry is hindered for more acidic substrates than the hydrogen donors. Such a phenomenon is rarely discussed before, and we also used DFT calculations to support this hypothesis. We believe that this understanding and discussion are highly attractive to readers, also guiding subsequent catalyst design for the transfer hydrogenation.

In conclusion, we believe that this study will attract extensive research interest from the readers of *Nature Communications* in catalytic activity, preparation routes, exploration of active site structures, methodological approaches to transition states, and guidance for constructing hydrogenation catalytic systems.